# TP-MDDN: Task-Preferenced Multi-Demand-Driven Navigation with Autonomous Decision-Making

**Shanshan Li[1], Da Huang[23], Yu He[13], Yanwei Fu[13†], Yu-Gang Jiang[1], Xiangyang Xue[1†]**
[1]Fudan University    [2]Shanghai Jiao Tong University
[3]Shanghai Innovation Institution

## Abstract

In daily life, people often move through spaces to find objects that meet their needs, posing a key challenge in embodied AI. Traditional Demand-Driven Navigation (DDN) handles one need at a time but does not reflect the complexity of real-world tasks involving multiple needs and personal choices. To bridge this gap, we introduce **Task-Preferenced Multi-Demand-Driven Navigation (TP-MDDN)**, a new benchmark for long-horizon navigation involving multiple sub-demands with explicit task preferences. To solve TP-MDDN, we propose **AWMSystem**, an autonomous decision-making system composed of three key modules: Break-LLM (instruction decomposition), LocateLLM (goal selection), and StatusMLLM (task monitoring). For spatial memory, we design MASMap, which combines 3D point cloud accumulation with 2D semantic mapping for accurate and efficient environmental understanding. Our Dual-Tempo action generation framework integrates zero-shot planning with policy-based fine control, and is further supported by an Adaptive Error Corrector that handles failure cases in real time. Experiments demonstrate that our approach outperforms state-of-the-art baselines in both perception accuracy and navigation robustness.

## 1  Introduction

In daily life, people often identify a need and look for something in their environment to meet it [3, 4]. Demand-Driven Navigation (DDN) [1] is a task where an agent receives a natural language instruction (e.g., "I am tired") and must find an object that fulfills that need (e.g., a bed or chair). This is a variation of the ObjectNav task. However, a single need can often be met in different ways, depending on personal preferences. For example, "organize the living space" could mean finding cleaning tools, decorative items, or storage boxes. To guide the agent effectively, it is important to clarify the user's specific preference, like focusing on decoration, so the instruction becomes actionable. People also tend to have a series of needs, such as cleaning, then resting, then eating. Efficiently handling multiple needs and evaluating the success of these actions is still a major challenge. This paper explores how to enhance long-horizon DDN tasks by making user preferences more explicit.

Recent work like MO-DDN [5] addresses navigation instructions through Multi-object Demand-driven Navigation with human preferences. For example, given an instruction like "I need to display my photography collection, preferably with good lighting", the demand may involve multiple objects such as picture frames, bookshelves, and ceiling lamps. However, it still focuses on single-demand navigation. Some works have explored long-range instruction navigation by constructing Landmark Semantic Memory for decision-making planning [6] or adopting Autonomous Evolution mechanisms [7, 8], but they primarily target object-driven navigation.

---

[†]Corresponding Authors

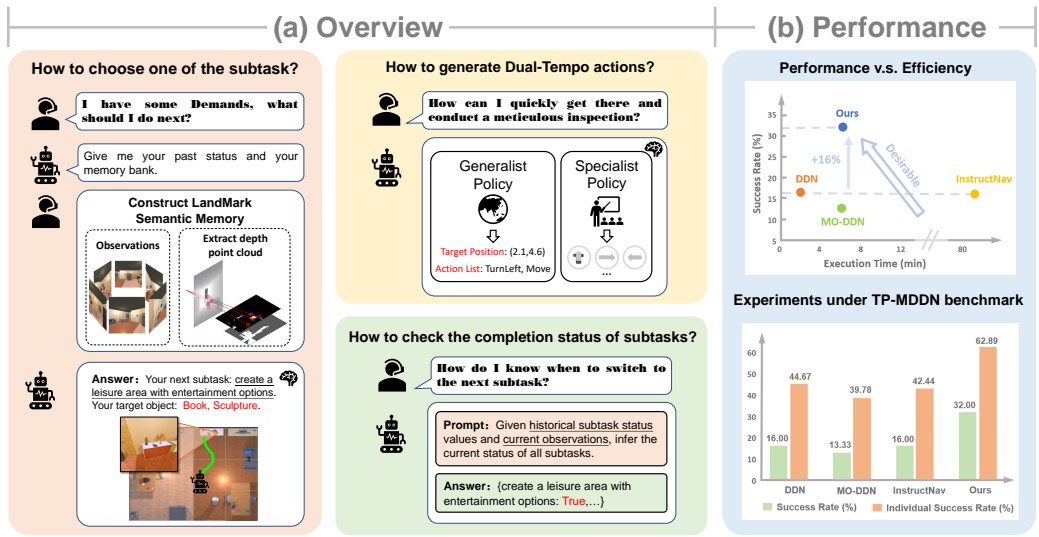

Figure 1: This figure presents our autonomous decision-making process and its performance benefits. (a) Overview: The pink area shows subtask selection using landmark semantic memory; the yellow area explains Dual-Tempo action generation via generalist and specialist policies; the green area details dynamic subtask completion checks. (b) Performance: Our method achieves a $16\%$ higher success rate than DDN [1] and InstructNav [2] under the TP-MDDN benchmark, with superior efficiency and individual success rates, highlighting its effectiveness and reliability.

We introduce a new benchmark called Task-Preferenced Multi-Demand-Driven Navigation (TP-MDDN) to handle scenarios with multiple needs, where each need includes a clear preference for a specific task-related object category. In a simulated multi-room home environment, we use DeepSeekV3 [9] and GPT-4o [10] to generate long-horizon instructions and object pairs across different scenes. These pairs are manually checked to ensure accuracy.

To efficiently manage memory in long-term navigation, we introduce a novel Multidimensional Accumulated Semantic Map (MASMap) that achieves a balance between accuracy and efficiency without requiring additional training. MASMap integrates local 3D point cloud accumulation with a global 2D semantic map to build and maintain spatial memory over time. A core challenge of merging semantically similar objects viewed from different perspectives is addressed using IoU-based fusion and Ram-Grounded-SAM for accurate segmentation and labeling. To reduce storage overhead, we implement an efficient update-and-prune strategy that preserves critical small objects. Our global map structure further minimizes redundancy while ensuring consistent and reliable semantic memory across extended navigation trajectories.

To address long-horizon visual navigation tasks, we further propose the **Autonomous Decision-Making World Model System (AWMSystem)**, inspired by WMNav [11]. AWMSystem breaks down complex instructions into sub-demands using **BreakLLM**, dynamically selects goals through **LocateLLM** based on object memory and execution status, and tracks task progress via **StatusMLLM** using real-time observations. For lightweight deployment, we employ a dual-tempo action generation strategy: zero-shot planning using A* algorithm with obstacle maps and affordance estimation (as in InstructNav [2]), and fine-grained policy-based control near targets, following DDN [1]. A major challenge in simulation environments, i.e., handling logical loops, boundary violations, and unseen obstacles, is addressed with an **Adaptive Error Corrector**, which adjusts actions in real time based on environmental feedback, greatly enhancing robustness. This modular design supports efficient, reliable long-term navigation without requiring additional end-to-end retraining.

We summarize our main contributions as follows: (1) We introduce **TP-MDDN**, a new long-horizon navigation benchmark with multi-sub-demand tasks and explicit task preferences, featuring high semantic richness and scene diversity through the use of DeepSeekV3 and GPT-4o. (2) We propose **AWMSystem**, an autonomous decision-making world model system composed of BreakLLM, LocateLLM, and StatusMLLM, enabling efficient instruction decomposition, dynamic goal selection,

and real-time execution monitoring without requiring end-to-end training. (3) We design a lightweight **MASMap**, which fuses 3D object detection and 2D semantic mapping to achieve both accurate perception and computationally efficient navigation. (4) Extensive experiments demonstrate that our method **significantly outperforms state-of-the-art baselines**, with ablation studies confirming the effectiveness of each component. Our system achieves strong robustness and environmental adaptability while maintaining low computational overhead, showcasing a practical balance between performance and efficiency that has not been addressed by previous works.

## 2 Related Works

**Vision-Language Navigation.** Visual-Language Navigation (VLN) involves guiding an agent to a goal based on language instructions and visual observations. Early methods focused on progress estimation [12, 13], backtracking [14], reinforcement learning [15, 16], and policy learning [1, 5, 17, 18, 19, 20, 21]. Some works extracted object and action types from instructions [22, 23, 24, 25, 26], while others leveraged transformers for history encoding [27, 21, 22, 23, 28], built topological maps [29, 20, 30], or predicted future events [31]. Pre-training [32, 33, 34, 35, 36] and data enhancement [37, 38, 39, 35, 40, 41] have also been explored. However, these approaches struggle with long-horizon continuous navigation. In contrast, our method adapts zero-shot scene layout understanding to achieve strong performance in such challenging tasks.

**Continuous VLN with Foundation Models.** Foundation models, like LLMs, VLMs, and LVLMs, have advanced visual navigation [42, 43] by enabling strong reasoning, high-level planning, and end-to-end action generation abilities. Recent zero-shot approaches use these models for collaboration [44], memory-based reasoning [45, 46], or instruction tuning [47]. However, most work focuses on discrete actions, while continuous navigation, more suited for real-world use, remains difficult [48, 1]. Earlier methods used GRU or LSTM [49, 50], while recent ones address object-targeted [51, 52, 53, 54] and instruction-following tasks [55, 56, 57]. Some predict progress [55], use value maps [58], or plan trajectories [59, 60, 61]. Yet, long-horizon tasks suffer from the high cost of frequent LLM calls. To solve this, we introduce a Dual-Tempo action generator, inspired by dual-system robotics [62, 63, 5], which boosts efficiency without losing performance.

**Long-Horizon Navigation.** Long-horizon navigation is essential for building agents that can learn and act over time. While benchmarks like LH-VLN [64] have made progress, success rates remain low due to limited memory encoding. Recent work has explored memory-based methods—like WMNav [11] for relation prediction and Mem2Ego [6] for using landmarks to guide decisions—but they focus on object goal navigation. Real-world systems like ReMEmbR [65] support long-range navigation but lack continuous control. Minecraft agents show promise through skill reuse [8, 7], but they rely on explicit object prompts. In contrast, demand-driven navigation, where agents meet high-level needs like "find an office tool", better reflects real scenarios. Yet, handling multi-step goals without object-specific instructions remains difficult. To tackle this, we combine obstacle avoidance, error correction, status tracking, and large foundation models to boost navigation performance in complex environments.

## 3 Tasked-Preferenced Multi-Demand-Driven Navigation

In a Task-Preferenced Multi-Demand-Driven Navigation task, the agent is given a natural language long-horizon instruction (e.g., *"Organize the living space by arranging decorative items, set up a cozy entertainment corner with seating and media devices"*), comprising multiple subtasks. Each subtask combines a basic requirement and a task preference; for example, the basic requirement is *"set up a cozy entertainment corner"*, and the task preference is *"with seating and media devices"*.

Formally, let $S$ denote the set of environments and $D$ denote the set of long-horizon instructions. For a given instruction $d = \langle d_1, d_2, \ldots, d_L \rangle \in D$, where $L$ is the number of subtasks and $d_i$ is the $i$-th subtask, the goal is to determine whether the agent successfully finds objects satisfying each subtask. Specifically, in each episode: (1) A random environment $s \in S$ is selected. (2) The agent is initialized at a random position and orientation within $s$. (3) A random instruction $d$ is chosen. (4) Let $O$ be the set of objects in the environment. We define a function $G : D \times O \rightarrow \{0, 1\}^L$, mapping instructions and environment objects to a binary vector indicating whether each subtask is satisfied. To complete the instruction, the agent must find at least one object for each subtask. Each subtask is

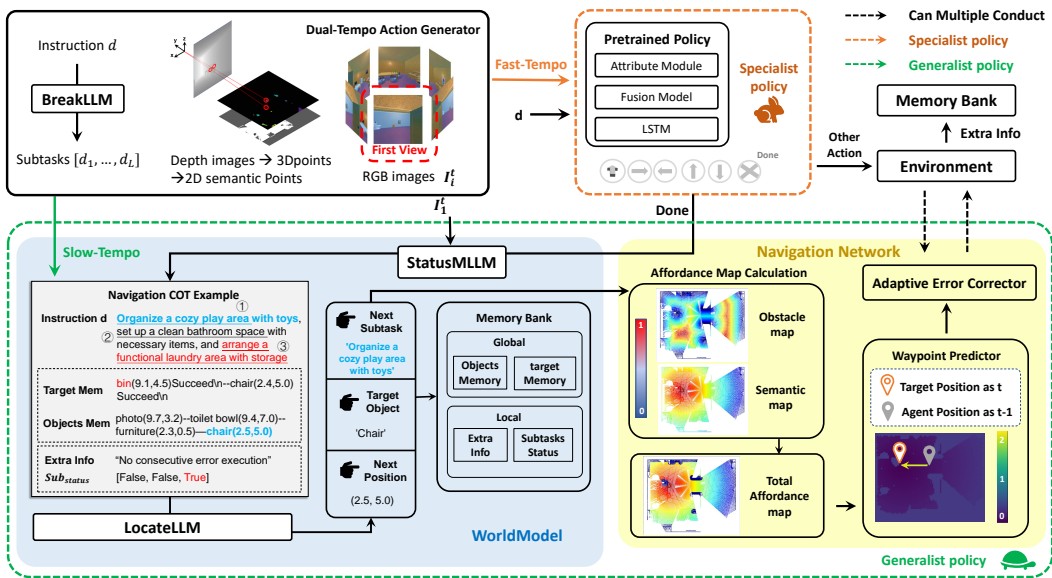

Figure 2: **Overview**. This diagram illustrates the dual-tempo action generation process in our system. The BreakLLM module decomposes the instruction. Then, depth images are converted into 2D semantic points. The fast-tempo branch uses a pretrained policy to generate primitive actions, while the slow-tempo branch employs LocateLLM for high-level navigation reasoning, determining target objects and positions. StatusMLLM tracks task progress and updates memory. The Navigation Network performs affordance map computation, adaptive error correction, and waypoint prediction.

strictly binary, defined as either success or failure. To reduce the difficulty of the TP-MDDN task, if the 2D Euclidean distance between the agent and the object is within a threshold $\epsilon_{\text{dis}}$, the object is considered found. This differs from DDN [1], which requires finding the target object within the field of vision.

At each step, the agent can generate actions using either the **Generalist Policy** or the **Specialist Policy**. The Generalist Policy drives the agent toward a target pose selected by the large model, while the Specialist Policy uses a pretrained network to execute one of six actions: `MoveAhead`, `RotateRight`, `RotateLeft`, `LookUp`, `LookDown`, and `Done`. The system invokes the Generalist Policy at regular intervals to explore potentially relevant regions, after which the pretrained policy [1] is used to search for objects matching the task requirements. The agent terminates the navigation task when the number of steps reaches the maximum length $Len_{\text{max}}$, or when all subtasks are completed. From the test set of ProcTHOR [66], we generated 200 long-horizon instructions, each containing three subtasks, spanning 68 rooms.

### 3.1 Autonomous Decision-Making World Model System

**AWMSystem Overview.** Figure 2 illustrates our Autonomous Decision-Making World Model System, which models the environment and predicts actions based on observations. It constructs a real-time 2D semantic map with efficient memory storage and uses past trajectories and layout information to select the next target. The Dual-Tempo action generator plans actions, while an Adaptive Error Corrector adjusts strategies based on feedback.

#### 3.1.1 Construction of Long-Range Memory Banks

**Raw Data Processing.** Building the memory bank consists of three components: input data processing, real-time accumulation, and recording of different types of historical data. For raw data processing, we perform object detection and segmentation on RGB images and extract 3D point clouds of detected objects from depth maps. These 3D points are then fused and compressed into a 2D semantic map. Specifically, at regular intervals, the agent performs environmental sensing. Suppose the current step is $t$; the agent captures $n$ RGB images $I_1^t, \ldots, I_n^t$ and corresponding depth

images $\text{Depth}_1^t, \ldots, \text{Depth}_n^t$. For each image $I_i^t$, we use the Ram-Grounded-SAM model [67, 68] to obtain object labels, bounding boxes, and segmentation masks. Next, we compute the real-world 3D point cloud $PC_{cur}$ from each depth image using the camera's intrinsic parameters and rotation matrix.

**Real-time Accumulation.** In the real-time accumulation design, since object point clouds obtained from different viewpoints at the current location may correspond to the same physical object, we design a point cloud update strategy. Let $r$ be an object point cloud in the recorded set $PC_R$, $r_{cur}$ be an object point cloud in $PC_{cur}$ and let $r_{cur}^{final}$ denote the residual point cloud obtained by removing regions of $r_{cur}$ that overlap with any object in $PC_R$. During this process, for each candidate $r \in PC_R$, we compute two overlap metrics based on the intermediate point cloud $r_{cur}^*$ at a given stage:

$$os^* = \frac{\text{overlap\_score}(r_{cur}^*, r)}{|r_{cur}^*.pcd|}, \quad ros^* = \frac{\text{overlap\_score}(r_{cur}^*, r)}{|r.pcd|},$$

where the overlap score is computed using element-wise Euclidean distance between point clouds and $|\cdot|$ denotes the number of points. Let $\text{Update}(r_{cur}, PC_R)$ denote the operation that updates the reference set $PC_R$ based on $r_{cur}$. We define it as follows:

$$\text{Update}(r_{cur}, PC_R) = \begin{cases} PC_R \cup \{r_{cur}\} & \text{if } \max_{r \in PC_R} os^* < 0.25 \\ r^*.pcd \leftarrow \text{Merge}(r.pcd, r_{cur}^{final}.pcd) & \\ r^*.class \leftarrow r_{cur}.class & \text{if } os^* > 0.8 \wedge ros^* > 0.8 \\ PC_R[r] \leftarrow r^* & \end{cases}$$

If the maximum $os^*$ over all $r \in PC_R$ is less than 0.25, $r_{cur}$ has negligible overlap with any existing object, so it is treated as a new object and added to $PC_R$. If both $os^* > 0.8$ and $ros^* > 0.8$ for a specific $r$, this indicates strong overlap, suggesting that $r_{cur}$ and $r$ represent the same object. In this case, their point clouds are merged, and the class label of $r$ is updated to that of $r_{cur}$.

**Fusion of Global Semantic Map.** After recording the center coordinates of object point clouds, all 3D point cloud data is cleared to save memory. Let $\mathcal{OM}_t$ denote the object memory bank storing information about previously detected objects up to time step $t$. Each object's 2D information is recorded as $\{\texttt{class} : c_{obj} \in \mathcal{C}, \texttt{center} : \mathbf{p_{obj}} = (x_{obj}^{center}, y_{obj}^{center}), \texttt{bbox} : [x_{obj}^{\min}, x_{obj}^{\max}, y_{obj}^{\min}, y_{obj}^{\max}]\}$, where $\mathcal{C}$ is the set of possible object classes. We compute the 2D IoU between current and historical objects and apply the Hungarian algorithm to find the most similar historical object. If a match exists, we update the corresponding entry in the 2D semantic map and object memory bank; otherwise, we add the new object. Under a task, we continuously accumulate the names and locations of explored objects to form a target memory. The data in the target memory bank is formatted as $\langle$Target Object, $(x, y)$, Feedback Type$\rangle$, where $(x, y)$ denotes the target location on the map, and *Feedback Type* includes: $\texttt{Success}$, $\texttt{Obstructed}$, $\texttt{Out-of-Bounds}$, and other failure descriptions.

**Explanation of Memory Bank.** As illustrated in Figure 2, the memory bank contains two components. Global Records serve as Long-Term Memory by storing visited information, including a cumulative map of detected objects with their 2D poses and a history of planned targets with execution outcomes, enabling continuous progress tracking. In contrast, Local Updates function as Short-Term Memory by maintaining transient data for immediate decision-making. This includes local 3D point clouds extracted from the current panoramic view, additional information derived from past execution failures to address recurring issues, and the current status of the ongoing subtask.

### 3.1.2 Summary of Foundation Model Usage

**BreakLLM.** We employ a Large Language Model (LLM) to automatically decompose long-horizon instructions into a subtask list $d_{sub}$, and initialize a corresponding subtask execution status list $Sub_{Status}$, which is maintained throughout the task. The instruction decomposition is formalized as $(d_{sub}, Sub_{Status}) = BreakLLM(d)$, where all entries in $Sub_{Status}$ are initially set to $\texttt{False}$.

**LocateLLM.** At time step $t$, we maintain a memory of detected objects and their 2D coordinates, denoted as $\mathcal{OM}_t$ (Object Memory). The primary inputs to the decision-making module include: the overall instruction $d$, the subtask list $d_{sub}$, the current subtask status $Sub_{Status}$, the target memory $\mathcal{T}_t$, and the object memory $\mathcal{OM}_t$. Large models may struggle to fully comprehend long sequences of historical trajectories, potentially leading to repeated failures on the same object. To mitigate this, we introduce auxiliary feedback to help the model detect and avoid execution

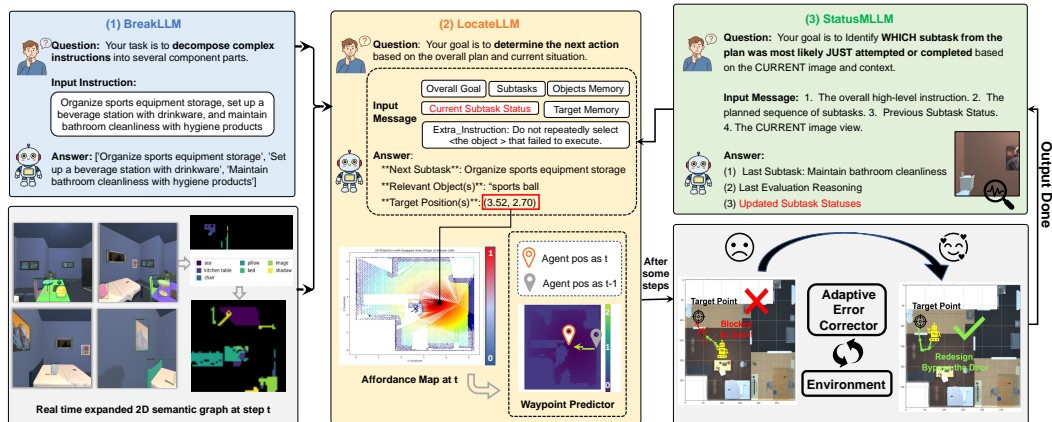

Figure 3: **Foundation Model Usage.** BreakLLM decomposes the instruction. The agent uses Ram-Grounded-Sam [67, 68] to segment panoramic RGB-D images and ultimately map them onto 2D semantic maps to form object memory. LocateLLM receives multiple types of data and outputs the next target object and position. StatusMLLM determines whether a subtask has been completed based on the current observed image. Adaptive error corrector re-plans the failed trajectory.

loops. Specifically, we track the number of consecutive failed attempts $n_{\text{CFE}}$ on the target object $TargetObj_{t-1}$. If $n_{\text{CFE}} \geq n_{\text{tolerance}}$, we generate an auxiliary prompt $Extra_{\text{info}}$ in the form: "Do not select `<object>` or `<position>` in the next step". This prompt is passed to the planner to discourage revisiting failed targets. The next target object is determined by:

$$TargetObj_t = \text{LocateLLM}\left(\mathcal{T}_t, \mathcal{OM}_t, Sub_{Status}, d, d_{sub}, \mathbb{I}\left(n_{\text{CFE}} \geq n_{\text{tolerance}}\right) \cdot Extra_{\text{info}}\right) \quad (1)$$

**StatusMLLM.** Iterative updating of subtask completion status is crucial for long-horizon task navigation. We introduce a multimodal LLM, StatusMLLM, to update $Sub_{Status}$ when the policy network outputs a Done action. This design is motivated by the observation that prior methods often trigger Done upon detecting an object that matches the instruction [21, 1]. We leverage this behavior to infer subtask completion. Specifically, we input the overall instruction $d$, the previous subtask status $Sub_{Status}^{t-1}$, the target memory $\mathcal{T}_t$, and the current image $I_1^t$ into StatusMLLM. The model outputs inference on which subtask might have just been completed $Sub_{cur}$, explanation for its judgment $Reason$, updated subtask completion status list $Sub_{status}^t$. This process is formalized as:

$$\left(Sub_{cur}, Reason, Sub_{Status}^t\right) = \text{StatusMLLM}\left(d, Sub_{Status}^{t-1}, \mathcal{T}_t, I_1^t\right) \cdot \mathbb{I}\left(A_{\text{policy}} = \text{Done}\right) \quad (2)$$

### 3.1.3 Dual-Tempo Action Generator

Inspired by prior work on two-stage navigation strategies [52, 11, 62, 63, 5] and considering that frequent invocation of large models at every step incurs significant computational overhead, we propose the **Dual-Tempo Action Generator**, as illustrated in Figure 2. This architecture decouples planning into a slow-tempo phase and a fast-tempo phase to balance reasoning depth with efficiency.

In the slow-tempo execution phase, we follow these steps: (1) Extract the current object point cloud $PC_{\text{cur}}$ from the panoramic observation and fuse it into the object memory $\mathcal{OM}_t$. (2) Feed historical context and execution feedback into LocateLLM to determine the next target object and its 2D location. (3) Compute affordance value maps that encodes navigational feasibility and semantic relevance. (4) Apply the A* algorithm on the affordance map to generate a globally feasible navigation path. (5) Decompose the path into a sequence of intermediate waypoints, convert each segment into discrete actions, and execute them sequentially. (6) Invoke the **Adaptive Error Corrector** to detect and rectify trajectory failures.

In the fast-tempo execution phase, we directly employ the pretrained policy from prior work [1], which outputs low-level actions based on the current RGB image and the high-level instruction. When the policy outputs Done, the StatusMLLM module is triggered to evaluate whether a subtask

has been completed. Next, we detail the computation of the affordance value map and the operational mechanism of the Adaptive Error Corrector.

**Calculation of Affordance Map**. At step $t$, we sequentially read an RGB image $I_i^t$ and a depth image $\text{Depth}_i^t$ from the panoramic views. For each depth image, we obtain its real-world 3D point cloud $PC_{\text{cur}}$. Based on the height of the points, we classify them into navigable points $\mathcal{N}_{\text{navi}}^{t,i}$ and obstacle points $\mathcal{O}^{t,i}$. Then, we project $\mathcal{N}_{\text{navi}}^{t,i}$ and $\mathcal{O}^{t,i}$ onto a 2D grid map to form $\mathcal{N}_{\text{grid}}^{t,i}$ and $\mathcal{O}_{\text{grid}}^{t,i}$, and perform dilation on the obstacle regions $\mathcal{O}_{\text{grid}}^{t,i}$.

Next, we construct the 2D affordance map by computing the obstacle avoidance affordance and semantic affordance. Following similar practices [2], we calculate the Euclidean distances from navigable points $\mathcal{N}_{\text{navi}}^{t,i}$ to category-specific point sets. For any navigable point $n_i \in \mathcal{N}_{\text{navi}}^{t,i}$, if its distance to the 2D obstacle point cloud satisfies $d_{\text{obs}}(n_i) < \tau_{\text{obs}}$, the obstacle avoidance affordance value $a_{\text{obs}}(n_i)$ is set to 0; otherwise, it is normalized as $a_{\text{obs}}(n_i) = (d_{\text{obs}}(n_i) - d_{\text{min}})/(d_{\text{max}} - d_{\text{min}})$. From the target 2D coordinates obtained in Section 3.1.2, we define the semantic affordance value $a_{\text{tgt}}(n_i)$ as the inverse of the normalized distance to the target point cloud: $a_{\text{tgt}}(n_i) = 1 - (d_{\text{tgt}}(n_i) - d'_{\text{min}})/(d'_{\text{max}} - d'_{\text{min}})$, where $d_{\text{tgt}}(n_i)$ represents the distance from $n_i$ to the target point. This means that positions closer to the target receive higher semantic affordance values. Here, $d_{\text{min}}$ and $d_{\text{max}}$ denote the minimum and maximum distances to the obstacle point cloud, respectively; $d'_{\text{min}}$ and $d'_{\text{max}}$ are the corresponding minimum and maximum distances to the target point cloud. Finally, we compute the final affordance value $a_{\text{final}}(n_i)$: if $a_{\text{obs}}(n_i) = 0$, then $a_{\text{final}}(n_i)$ is set to 0; otherwise, it is set to the clipped value of $a_{\text{tgt}}(n_i)$ between 0.1 and 1. The formula is as follows:

$$a_{\text{final}}(n_i) = \begin{cases} 0, & a_{\text{obs}}(n_i) = 0 \\ \text{clip}\left(a_{\text{tgt}}(n_i),\ 0.1,\ 1\right), & \text{otherwise} \end{cases} \tag{3}$$

**Adaptive Error Corrector.** Our Adaptive Error Corrector uses environmental feedback to correct navigation errors. When the agent detects that a `MoveAhead` action may lead to a collision (e.g., with doors, walls, or furniture), it re-plans a new path from the current position and updates the waypoint sampling strategy. Under normal operation, the agent navigates by sampling a waypoint every $n_{\text{waypoint}}$ steps, using a discrete action space defined by forward translations of 0.25 meters and rotational increments of 30 degrees. When re-planning is triggered due to execution failure, the agent continues to operate within the same action space. The affordance map is recomputed based on the current state, allowing the planner to generate a revised trajectory that avoids obstacles and resumes progress toward the target object. To improve navigation precision, the trajectory is divided into two segments: an initial segment and a subsequent segment. In the initial segment, a finer sampling interval $n_{\text{block}}$ is applied to support detailed spatial reasoning near obstacles. In the subsequent segment, the sampling frequency reverts to the standard rate $n_{\text{waypoint}}$, consistent with the regular navigation strategy.

# 4 Experiments

## 4.1 Experimental Setups

**Experiments.** We use AI2-THOR [69] as our simulator and ProcThor as our scene dataset [66]. We used each of DeepSeek-V3 [9] and GPT-4o [10] to generate 100 task-preferenced, multi-demand-driven unseen instructions in test scenarios (totaling 200 commands). In all experimental settings, the success distance threshold $\epsilon_{\text{dis}}$ is 1.5 meters, the maximum step count $Len_{\text{max}}$ is 50, and the tolerance for repeated failed attempts on the same object $n_{\text{tolerance}}$ within $Extra_{\text{info}}$ is 2. The obstacle avoidance distance $\tau_{\text{obs}}$ used during affordance map computation is 0.25 meters. When processing input data, we set the camera resolution to 300×300 and the horizontal field of view (HFoV) to 90° for the agent. The agent operates in a closed-loop fashion, perceiving environmental feedback (e.g., success, collision, or boundary detection) immediately after executing each discrete action. All experiments can be run on a single NVIDIA H100 80GB GPU.

**Evaluation Metrics.** In line with prior works [1, 5, 64], we adopt the following evaluation metrics: (1) Success Rate (SR): The proportion of tasks in which the agent successfully reaches the target object associated with each subtask. (2) Independent Success weighted by Path Length (ISPL): For each task, the success of each subtask is weighted by the ratio of the shortest path length to the

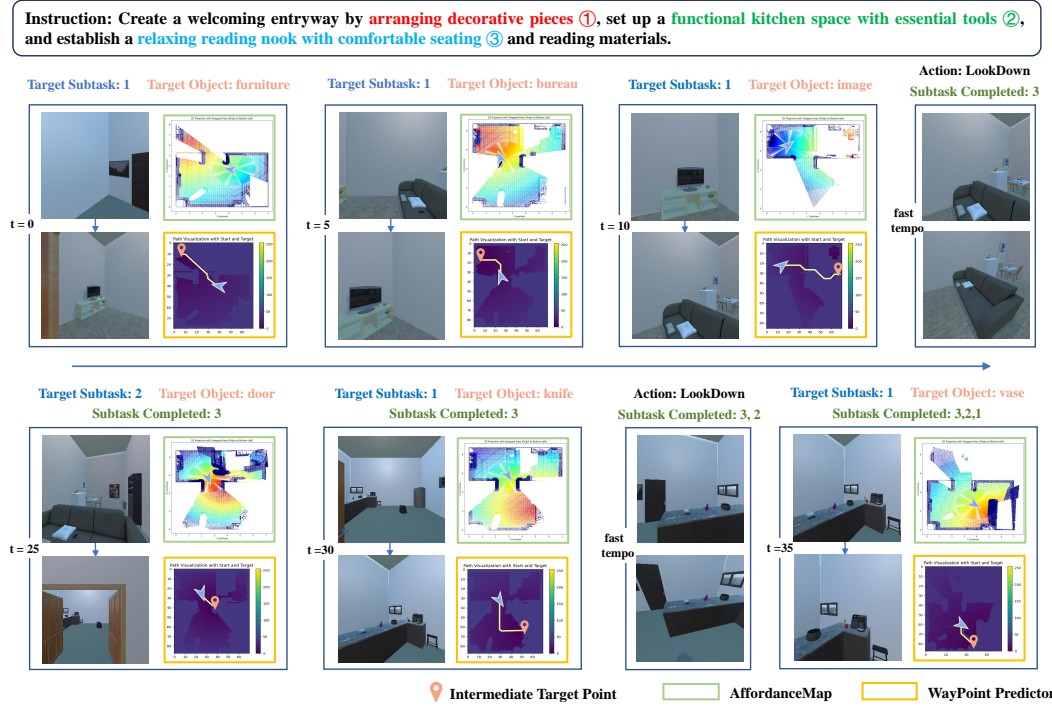

Figure 4: **Visualization Results**. The intelligent agent receives a Task-Preferenced Multi-Demand-Driven instruction, autonomously decomposes it into multiple subtasks, and identifies objects in the scene that match the unexecuted subtasks. On the affordance maps, redder values indicate higher affordance scores. The arrow in the waypoint predictor graph represents the agent's location and current field of view. As the step count increases, the three subtasks are gradually completed.

actual path length, then averaged across all subtasks. (3) Successful Trajectory Length (STL): The average number of steps taken in successful navigation trials. (4) Independent Success Rate (ISR): The success rate for each subtask evaluated individually. We evaluate the above metrics over 50 tasks for each method, averaging three runs due to the randomness in large model outputs.

**Baselines**. DDN is an end-to-end, single-demand-driven navigation method [1]. MO-DDN is an attribute-based exploration modular agent designed for multi-object, demand-driven navigation [5]. InstructNav uses dynamic chain-of-navigation and multi-sourced value maps to generate robot-actionable trajectories [2]. These three methods are the most relevant and state-of-the-art in the field of demand-driven navigation.

## 4.2 Main Results

The experimental results with baselines are shown in Table 1. Our AWM-Nav achieves the highest scores across all metrics, demonstrating its superior performance in long-horizon navigation tasks. In contrast, all baseline methods achieve significantly lower results than ours, which can be attributed to the fact that long-horizon instruction navigation typically requires stronger active exploration capabilities and the ability to execute multiple subtasks, while these baselines are designed for single-task navigation and lack such capabilities. InstructNav [2] combines zero-shot reasoning using large models with explicit memory mechanisms. However, its planned paths often result in collisions within the environment. Although InstructNav employs dynamic chain-of-thought reasoning over actions, the LLM struggles to infer the completion status of subtasks from raw action sequences due to limited contextual understanding. MO-DDN [5] adopts a two-stage navigation process consisting of coarse search followed by fine-grained localization. As the full implementation of MO-DDN has not been publicly released, we instead integrate its policy network with our own MASMap for the coarse exploration stage, achieving the lowest success rate. Despite showing some capability in multimodal alignment, MO-DDN still falls short when coping with the challenges posed by our

Table 1: Comparison with state-of-the-art methods on the TP-MDDN benchmark. In the Large-model Inference column, ✓ indicates the LLM is used for reasoning. "Explicit History" refers to methods that record object names and positions in the scene.

| Method | Zero-shot | Large-model Inference | Explicit History | STL↑ | ISR↑ | SR↑ | ISPL↑ |
|--------|-----------|----------------------|------------------|------|------|-----|-------|
| DDN [1] | ✗ | ✗ | ✗ | 15.50 | 44.67 | 16.00 | 40.66 |
| MO-DDN [5] | ✗ | ✓ | ✓ | 12.11 | 39.78 | 13.33 | 36.25 |
| InstructNav [2] | ✓ | ✓ | ✓ | 9.50 | 42.44 | 16.00 | 39.41 |
| **AWM-Nav** | ✗ | ✓ | ✓ | **20.11** | **62.89** | **32.00** | **44.19** |

Table 2: Ablation results for object segmenters, reasoning large models, Adaptive Error Corrector, and StatusMLLM. **-** means no information in this line, ✓ means using the method, and ✗ means not using the method.

| 1. The effect of different object segmenters | | | | | 2. Different reasoning large models | | | | |
|--------|------|------|------|------|--------|------|------|------|------|
| Method | STL↑ | ISR↑ | SR↑ | ISPL↑ | Method | STL↑ | ISR↑ | SR↑ | ISPL↑ |
| GLEE [70] | 14.94 | 51.11 | 21.33 | 41.05 | Qwen2.5-VL-7B | 10.97 | 47.78 | 19.33 | 36.45 |
| YOLO | 15.56 | 58.00 | 29.33 | 43.69 | GPT-4o | 17.51 | 56.44 | 28.67 | 39.95 |
| **RAM-Grounded-SAM [67, 68]** | **20.11** | **62.89** | **32.00** | **44.19** | **Qwen2-5-VL-72B** | **20.11** | **62.89** | **32.00** | **44.19** |
| 3. Influence of Adaptive Error Corrector | | | | | 4. The effect of StatusMLLM | | | | |
| BlockCorret | BeyondCorret | STL↑ | ISR↑ | SR↑ | ISPL↑ | With/Without | STL↑ | ISR↑ | SR↑ | ISPL↑ |
| ✗ | ✓ | 13.49 | 59.33 | 27.33 | 43.25 | - | - | - | - | - |
| ✓ | ✗ | 16.86 | 60.44 | 28.00 | 42.20 | ✗ | 15.94 | 60.67 | 27.33 | 42.46 |
| ✓ | ✓ | **20.11** | **62.89** | **32.00** | **44.19** | ✓ | **20.11** | **62.89** | **32.00** | **44.19** |

task-preferenced, multi-demand navigation scenarios. Note that STL refers to the average length of successful trajectories, and our higher score is due to solving some long-distance tasks that require crossing rooms, which increases the average trajectory length.

Besides the metrics in this table, we also pay particular attention to the average execution time, shown in Figure 1. In continuous-action navigation, using LLM inference at every step can be very time-consuming, as demonstrated by InstructNav's performance in (b). The average execution time per long-horizon instruction is 6.82 minutes for AWM-Nav, 1.74 minutes for DDN [1], 6.79 minutes for MO-DDN [5], and 88.90 minutes for InstructNav [2]. In the detailed time breakdown, slow-paced actions account for approximately 22 times the duration allocated to fast-paced actions in our method. In summary, our method adopts a dual-tempo action generator to save time and uses an automatic decision-making system built with large models to enhance the agent's reasoning capability.

**Ablation Studies**. As shown in Table 2, regarding the effect of **different object segmenters**, the RAM-Grounded-SAM-based [67, 68] model achieves the best performance, while GLEE [70] lacks precision and YOLO (Ultralytics YOLOv11) performs suboptimally. For different **reasoning large models**, the well-known GPT-4o does not lead to significant improvements, possibly due to the strong context understanding and state switching awareness required in long-horizon navigation tasks. The open-source Qwen2-5-VL-72B [71, 72] achieves the best metrics. After examining the execution behavior of the agent, it was found that the number of parameters in the large model affects the performance of intelligent planning. In evaluating the influence of the **Adaptive Error Corrector**, replanning affordance maps proves effective, and unexpected situations remain important to monitor and avoid, even with strong large model reasoning capabilities. With regard to **the effect of StatusMLLM**, task status tracking is crucial for long-horizon instructions, as misjudgment or absence of status reasoning can severely impact the success of the entire trajectory.

## 5 Conclusion and Discussion

This paper introduces a new benchmark, TP-MDDN, to address navigation tasks involving multiple sub-demands and explicit task preferences. Meanwhile, it proposes the AWMSystem decision-making system, MASMap spatial memory scheme, Dual-Tempo action generation framework, and an adaptive error corrector, which effectively tackle the challenges in TP-MDDN. Experiments show that the method achieves higher navigation accuracy than existing baselines and offers faster inference speed. However, the method has issues such as involuntary mode switching in the dual-tempo action generation framework and navigation decision errors caused by instruction misjudgment due to over-reliance on pre-trained large language models. Future work includes optimizing the mode switching

of the action generation framework through reinforcement learning and training domain-specific language models to reduce dependence on pre-trained models.

## 6 Acknowledgments

This work is supported in part by NSFC Project (No. 62176061) and Joint Laboratory of Intelligent Construction Engineering Technology for Operating Railway Lines, and the Science and Technology Commission of Shanghai Municipality (No. 24511103100). The authors gratefully thank these organizations for their support and resources.

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
