# OpenReview forum: "TP-MDDN: Task-Preferenced Multi-Demand-Driven Navigation with Autonomous Decision-Making"
_NeurIPS.cc/2025/Conference — NeurIPS 2025 poster_

### Official Review · Reviewer_7Quy · 2025-06-23

**Clarity:** 2
**Significance:** 3
**Originality:** 3
**Rating:** 4
**Confidence:** 4

**Summary:**

This paper introduces TP-MDDN, a new benchmark for long-horizon embodied AI navigation that handles multiple sub-demands with explicit task preferences, reflecting real-world complexity where people have multiple related needs. The proposed AWMSystem, an autonomous decision-making system, is composed of BreakLLM (instruction decomposition), LocateLLM (goal selection), and StatusLLM (task monitoring). The framework is combined with a MASMap for efficient 3D-to-2D spatial memory and a Dual-Tempo action generation framework that integrates zero-shot planning with policy-based fine control. The approach also includes an Adaptive Error Corrector for real-time failure handling. Empirical results demonstrate superior performance of TP-MDDN across all metrics while maintaining computational efficiency compared to methods using LLMs at every step. The system successfully balances performance and efficiency for practical long-horizon navigation tasks.

**Questions:**

Please consider answering the following technical questions for clarification.
- The authors mentioned in L206 that the output of slow-tempo is a sequence of continuous actions, but the output for fast-tempo/specialist policy are six discrete actions (as stated in L125 - L126), is that correct? If so, why is there such a discrepancy? Moreover, if the six actions for specialist policy are discrete, then the details about distance for MoveAhead, degrees for rotation and look actions are missing,
- In Figure 2, the Adaptive Error Corrector is placed inside the module of slow-tempo, where the actions should be sequence of continuous ones. However, in L235, you mentioned the agent senses feedback of some discrete action (MoveAhead), which shows contradiction.
- Still in L206, according to the authors, the sequence of actions are executed after decomposed from the navigation path. Does this mean the actions are executed in an open-loop manner? Does the agent gets feedback from the environment during this execution? If not, then there might be the situation where a collision or out-of-bound has occurred before the action sequence is completely executed.
- Could the authors provide more details regarding re-planning, how it exactly "starts with a finer interval  then gradually returns to the
usual"?
- The questions regarding clarity in Weaknesses.

**Ethical Concerns:**

["NO or VERY MINOR ethics concerns only"]

**Final Justification:**

The submission is generally good but with some clarity issues. The authors clarified multiple technical details and addressed concerns regarding this matter. Their detailed elaboration on the replanning mechanism is particularly appreciated. Therefore, I am inclined to accept this paper.

**Limitations:**

Yes

**Paper Formatting Concerns:**

The paragraph between L150 and L151 misses line numbers.

**Quality:**

3

**Strengths And Weaknesses:**

Strengths
- The introduced TP-MDDN benchmark advances beyond existing multi-object navigation by explicitly incorporating task preferences alongside multiple sub-demands, which addresses a real limitation where previous work, e.g., MO-DDN, would introduce ambiguous specification. Instead, the formalization of explicit task preferences better reflects real-world navigation complexity and provides clearer guidance for agent behavior.
- The systematic decomposition into three specialized LLM modules represents a reasonable and novel approach to handling complex long-horizon tasks. Each module addresses a distinct aspect of the navigation problem, which is both technically sound and aligns well with how humans might approach such tasks. The modular design allows for targeted evaluation and optimization.
- The Dual-Tempo action generation framework addresses a fundamental computational efficiency problem in LLM-based navigation by separating slow-tempo LLM reasoning for high-level planning and target selection from fast-tempo policy-based control for fine-grained actions near targets. Such design avoids the computational bottleneck of calling expensive LLMs at every step while maintaining the benefits of LLM reasoning for complex decision-making.

Weaknesses
- Regarding long-horizon instructions generation and split, according to the example illustrated in Fig. 3, the instructions are well-organized with straightforward sentence segmentation, where the BreakLLM simply breaks it into three parts split by comas. This is no more than directly feeding subtasks into the subsequent module. In the real-world scenario, a human instruction can be less clear and structured, although it may still clearly state the preference. Why not make the instructions more similar to human orders and take the advantage of LLM’s ability of language understanding? Moreover, the subtask completion status are uniformly set to False in BreakLLM. All these factors make the BreakLLM module trivial.
- [L161] It is unclear what global records and local updates are, respectively.
- There is no explicit reference to Eq. (1) and Eq. (2). Despite relevant, with out references, the two equations look independent of the text, especially for Eq. (1), where $H_t$, $M_t$, $State_t$ are never defined and explained in the text.
- [L181] The design of including extra info prompt after failing consecutive times is not that elegant. Instead of blaming large model’s capability of understanding long historical sequences and using an enforced way to circumvent it, it would be better to dig into the failed cases and analyze the culprit more carefully, providing a more neat solution to avoiding the failure beforehand.

---

> ### Author Rebuttal · Authors · 2025-07-31
>
> # Response to Reviewer 7Quy
>
> We sincerely thank the reviewer for their thorough review and insightful feedback.
>
> ## Response to Weakness 1:  Instruction Generation and BreakLLM's Role
>
> - We appreciate the reviewer's feedback regarding our instruction set design. The core purpose of BreakLLM is to decompose abstract human instructions into executable subtasks. Our current instructions are structured to ensure that each subtask's requirements are clear and unambiguous, which is essential for accurately evaluating the agent's understanding and execution, and helps prevent failures caused by ambiguous descriptions. We strongly agree that enhancing realism is important and are considering adopting a more natural instruction format. However, these instructions will still only convey basic needs and preferences without explicitly naming objects, in order to preserve the core challenge of the task—consistent with the instruction set used in our main article. Thank you very much for your suggestion.
> - An example of more colloquial instructions is as follows: "Before the guests arrive, please organize the sports equipment using some storage tools. Also, I'd like to set up a small beverage station, which requires preparing some drinkware, and please do a quick clean of the bathrooms with hygiene products to ensure everything looks good." We believe BreakLLM remains effective under such instructions.
> - The uniform initialization of all subtask completion statuses to False in BreakLLM is intentional, as it accurately reflects the logical starting point: no subtasks have been completed immediately after parsing a long-horizon instruction. This procedure provides a clear initial value for subsequent modules (e.g. StatusLLM) to accurately monitor and update progress.
>
> ## Response to Weakness 4:  Design for Handling Consecutive Failures
>
> We thank the reviewer for the insightful suggestion. While our current approach effectively mitigates immediate issues, we acknowledge that it can be further improved and extended with a more in-depth analysis.
>
> 1. LLM Base Impact on Planning: The choice of large language model (LLM) significantly influences planning quality. In our experiments, Qwen3-32B generated fewer repeated object predictions than Qwen2.5-VL-7B under identical conditions—using the same segmentation module and prompts. This suggests that the base model’s reasoning capabilities directly affect planning reliability, particularly in processing complex multimodal interaction histories. Our findings underscore the importance of advancing LLM backbone performance for instruction-following navigation tasks.
>
> 2. Cause of Consecutive Failures and Our Design Inspiration: A typical cause of repeated failures arises when targeting large objects (e.g., a sofa). The LocateLLM predicts the object's center, and the Waypoint Predictor generates a trajectory ending within it. If the agent fails to reach this point, the LLM may re-select the same object as the next target, creating a failure loop. Since precise 3D object sizing remains challenging, we instead leverage auxiliary information to break this cycle—demonstrating it to be an effective and concise solution.
>
> 3. Future Work Toward a More Elegant Solution: We agree that a more elegant approach is desirable. Our future work will explore adaptive trajectory planning using object size estimates derived from the 2D map, generating approach points outside the object's boundary based on its dimensions. This aims to enable robust navigation without relying on ad hoc fixes.
>
> ## Response to Questions
>
> ### **Response to Questions 1**:
>
> - Thank you for the feedback. We apologize for any confusion caused by the wording in L206 and will correct it. To clarify, both the slow-tempo and fast-tempo policies use discrete actions in our simulation environment, where fundamental actions like MoveAhead, RotateRight, and Lookup are inherently discrete. The difference lies in their execution: (1) the fast-tempo policy (Lines 125–126) executes individual discrete actions—e.g., a single 0.25m MoveAhead or 30° rotation step; (2) the slow-tempo policy (L206) refers to the outcome of a sequence of discrete actions, where a planned trajectory is broken into waypoints, each reached via multiple discrete steps. Slow-tempo movements use 0.25m steps and 45° rotations, chosen to balance precision and efficiency.
>
> ### **Response to Questions 2**:
>
> - Thank you for pointing out the potential source of confusion regarding Figure 2 and line 235. To clarify: (1) The adaptive error corrector is triggered when the fast-tempo process fails to execute a specific discrete action toward a given waypoint. Therefore, the environmental feedback it receives corresponds to discrete actions such as MoveAhead. (2) Furthermore, when the adaptive error corrector encounters issues like collisions, it invokes the waypoint predictor to replan the affordance map and fine-tunes the action interval design. Since this mechanism is closely tied to the planning strategy used in the fast-tempo process, it is logically situated within the fast-tempo framework.
>
> ### **Response to Questions 3:**
>
> - We appreciate this important question. Our agent operates in a closed-loop manner: after each discrete action, it receives immediate environmental feedback (e.g., success or collision or boundary detection). If an action fails, the system quickly adjusts the strategy, enabling robust navigation—this is key to the effectiveness of components like the Adaptive Error Corrector. The same feedback is available to other researchers using the environment and the difference lies in whether they choose to leverage it effectively.
>
> ### **Response to Questions 4: Details on Re-planning Mechanism**
>
>  We appreciate the opportunity to elaborate on our re-planning mechanism:
>
> 1. Re-planning Triggers: Failures typically arise from collisions during forward movement (e.g., in narrow passages, with table corners, or partially open doors) or boundary violations. When detected, re-planning is initiated to navigate around the obstacle.
>
> 2. Need for Refined Re-planning: We have attempted to input current observation images and context to directly prompt the LLM to generate a sequence of actions to resolve the current obstruction situation. However, experiments showed that renowned large models such as DeepSeek, GPT-4o, and Qwen2.5VL-72B were unable to generate effective action sequences, possibly due to a lack of experience in navigation trajectory planning within simulated environments. This motivated our geometric re-planning strategy.
>
> 3. Refined Re-planning Approach. The approach consists of three key components: First, a new affordance graph is generated for the agent's immediate surroundings, enabling a local update of the environment representation. Second, the central coordinate of the original target object remains unchanged, ensuring continuity in the goal. The core innovation lies in the **variable sampling interval**: when re-planning the trajectory, sampling points are placed at smaller intervals near the beginning (head) of the trajectory, while maintaining the original sampling interval for the latter portion. Experimental results demonstrate that this method effectively helps agents navigate around obstacles and significantly improves the overall task success rate.
>
> 4. Validation \& Example: To better understand the Adaptive Error Corrector section (illustrated in the lower right corner of Figure 3 in the main text), please refer to the animation task in the far left panel of '**simulation-robot-demo.mp4**' (included in the attachments) at 0:01:08. The task instruction is "**Organize sports equipment storage, set up a beverage station with drinkware, and maintain bathroom cleanliness with hygiene products**". At t=34 seconds, the agent detects the target object in the bathroom within the panoramic view (note that not all panoramic views observed by the agent are shown in the animation). The agent initially plans a trajectory toward the target, but this path is blocked by a closed door. Upon detecting the collision, the system recognizes the action failure and triggers a replanning process. From the agent's current position, a new trajectory to the target object is generated. Thanks to a very small sampling interval for the head trajectory, the agent is able to first execute a turn maneuver, successfully navigating around the obstacle.
>
> ## Clarity Questions from Weaknesses Section
>
> ### **Response to Weaknesses 2**:
>
> Global Records (Long-Term Memory) stores visited information: a cumulative map of detected objects and their 2D poses, and a history of planned targets with their execution outcomes (success/failure), enabling progress tracking. Local Updates (Short-Term Memory) holds transient data for immediate decisions: local 3D point clouds extracted from the current panoramic view (8 perspectives), environmental feedback (e.g., from repeated execution failures), and the current subtask status.
>
> ### **Response to Weaknesses 3**:
>
> We sincerely thank the reviewer for pointing out this issue and we will revise this in the later version as follows: Equation (2), which formalizes the process described in Lines 185-194, will be explicitly referenced at that location. For Equation (1), corresponding to Lines 175-177, we will clearly define all variables correctly within the equation: $H_t$ (Historical Exploration Information, revised to $T_t$), $M_t$ (object memory bank, revised to $OM_t$), $State_t$ (current subtask list, revised to $Sub_{status}$), and $d_{sub}$ (subtask details, revised to $Subtasks$). These revisions will ensure that all symbols are properly introduced and the equations are fully understandable.

---

> > ### Comment · Reviewer_7Quy · 2025-08-02
> >
> > I thank the authors for their high-quality rebuttal, which clearly addressed my concerns and questions. Specifically, the authors clarified multiple technical details that hadn't been clearly stated in the main paper. Also, their detailed elaboration on the replanning mechanism is particularly appreciated.
> >
> > I have a few further question for the authors regarding the **variable sampling interval**. What is your specific way of controlling the granularity of intervals, i.e., $n_{\text{block}}$ and $n_{\text{waypoint}}$? Do you manually select values for them, or is there any automatic selection technique? Are there only two interval sizes, where after certain steps of $n_{\text{block}}$, it immediately switches to $n_{\text{waypoint}}$, or is it a smooth process? Finally, although the authors already claimed that "this method effectively helps agents navigate around obstacles", will the denser waypoints require more precise actions? For example, you have 0.25m for MoveAhead and 30° for rotation, are they smoothly applied to smaller intervals?
> >
> > I look forward the authors' response.

---

> > > ### Author Response · Authors · 2025-08-04
> > > **Questions about Variable Sampling Interval**
> > >
> > > **Response to Reviewer 7Quy**
> > >
> > > We sincerely thank the reviewer for the kind words and thoughtful feedback on our rebuttal.
> > >
> > > ### **1. Automatic or Manual Selection of Sampling Intervals**
> > >
> > > We currently use manual assignment, carefully designed based on the agent’s step size, visual range, and planning grid resolution.
> > >
> > > ### **2. Interval Sizes and Switching Strategy**
> > >
> > > Let the waypoints from the current location to the target be $[v_0, \dots, v_K]$, with $n_{\text{waypoint}}$ denoting the interval without replanning. After replanning, the new waypoints are $[v_0, \dots, v_{\text{head}}, v_{\text{head}+1}, \dots, v_N]$. The first segment $[v_0, \dots, v_{\text{head}}]$ uses a small interval $n_{\text{block}}^{\text{Head}}$, while the second segment $[v_{\text{head}+1}, \dots, v_N]$ uses a large interval $n_{\text{block}}^{\text{After}}$. The switch from $n_{\text{block}}^{\text{Head}}$ to $n_{\text{block}}^{\text{After}}$ occurs immediately after $v_{\text{head}}$.
> > >
> > > ### **3. Action Execution with Denser Waypoints**
> > >
> > > When replanning is triggered, we maintain the same discrete action standards: 0.25m for $\texttt{MoveAhead}$ and 30$^\circ$ for rotation.
> > >
> > > ### **4. Rationale and Design Considerations**
> > >
> > > Our navigation pipeline projects 3D points into a 2D plane, voxelizes them into a grid map, plans a trajectory between current and goal grids, samples waypoints, and computes discrete actions toward each next waypoint.
> > >
> > > - **Pre-replan period**: Due to a blind zone around the agent (no 3D points within a certain radius), sampling starts from a specific grid on the trajectory. Since diagonal movements cover a longer distance (approximately $\sqrt{2}$ grid units) compared to straight movements (1 grid unit), we set the waypoint interval $n_{\text{waypoint}}$ (e.g., 7 grids) to approximate the spatial extent of one diagonal step, ensuring consistent coverage across both movement types.
> > >
> > > - **Post-replan period**: Upon obstacle detection, the agent replans. The initial segment $[v_0, \dots, v_{\text{head}}]$ contains critical turning information. The interval $n_{\text{block}}^{\text{After}}$ is larger than $n_{\text{waypoint}}$, as the agent rarely encounters consecutive obstacles when targeting an object within the panoramic view.
> > >
> > > - **Action execution for fine intervals**: For $[v_0, \dots, v_{\text{head}}]$, where intervals are smaller than one step, oscillations may occur (e.g., forward then back). Two outcomes are possible: (1) the agent successfully bypasses the obstacle, or (2) it oscillates and returns to a position close to where replanning was triggered. Before analyzing the second case, we explain how the discrete action sequence is generated from the target waypoint. We compute multiple rotation actions to keep the angle between the agent's current view and the direction to the target within [-30°, 30°], and then determine the $\texttt{MoveAhead}$ action. In case (2), since the first waypoint $v_{\text{head}+1}$ after the initial segment is typically placed just beyond the obstacle, the agent will, after several rotations, align itself toward the correct direction and move forward.
> > >
> > > - **Transferability to sim2real or new environments**: Parameters $v_{\text{head}}$, $n_{\text{block}}^{\text{Head}}$, $n_{\text{waypoint}}$, and $n_{\text{block}}^{\text{After}}$ were determined through extensive exploratory experimentation. We have provided guidance for new environments or real-world deployments, where similar parameter tuning is required based on agent step size, visual range, grid resolution, obstacle-blocking distance, and post-turn distance. As a fallback, if the agent fails to reach the current target, the LLM will select a new object as the navigation goal.
> > >
> > > - **Future work on higher-precision replanning**: Higher accuracy requires finer grid maps and identification of key turning points. However, trajectories often contain many minor turns, making key-point detection challenging. Future work could segment trajectories into polyline segments at critical turns, guiding the agent along each segment. This is necessary because, although the agent can decide movement distance autonomously, current tasks still rely on discrete actions (e.g., $\texttt{MoveAhead}$, $\texttt{TurnLeft}$, $\texttt{TurnRight}$), and collision avoidance must still be ensured. While this involves engineering refinement, our focus remains on solving the core innovation of multi-demand navigation.
> > >
> > > Once again, we deeply appreciate the reviewer’s perceptive questions and constructive engagement with our work. Should you have any further questions, we would be more than happy to provide additional clarification.

---

> > > > ### Comment · Reviewer_7Quy · 2025-08-05
> > > >
> > > > I appreciate the authors' further explanation. Thus, my main concerns have been mostly addressed. I am happy to maintain my positive rating. I hope the authors include necessary details during this discussion in their final version to ensure an accurate and clear elaboration.

---

> > > > > ### Author Response · Authors · 2025-08-08
> > > > >
> > > > > ## **Response to Reviewer 7Quy**
> > > > >
> > > > > - We are deeply grateful to the reviewer for the thoughtful and diligent efforts throughout the review process, and for maintaining a positive rating. The insightful comments have greatly helped us improve the rigor and clarity of the manuscript.
> > > > >
> > > > > - As suggested, we will include the necessary details discussed during the review process in the final version to ensure an accurate and clear elaboration, including the explanation of instruction generation, the design rationale for handling consecutive failures, the decision-making process regarding fast and slow rhythms, the re-planning mechanism, as well as other clarifications.
> > > > >
> > > > > - We sincerely thank the reviewer once again for the time, consideration, and valuable feedback.

---

### Official Review · Reviewer_gPWR · 2025-06-30

**Clarity:** 3
**Significance:** 3
**Originality:** 1
**Rating:** 4
**Confidence:** 2

**Summary:**

The paper introduces TP-MDDN, a benchmark addressing task-preferenced, multi-demand-driven long-horizon navigation scenarios. It proposes an integrated solution, AWMSystem, combining several distinct components: BreakLLM for decomposing complex instructions into subtasks, LocateLLM for goal selection, StatusLLM for monitoring task progress, MASMap for spatial memory management, and a Dual-Tempo action generation framework supported by an Adaptive Error Corrector to handle navigation errors dynamically.

**Questions:**

1. Could the authors clearly differentiate between operational "preferences" and simple subtask criteria or object categories?
2. Is it possible to partially complete subtasks, or are subtasks strictly binary (fully successful or failed)? Clarifying this operational detail would significantly aid in understanding the task complexity.
3. Could the authors provide more explicit analysis regarding computation time versus robot traversal time?
4. Why are traditional temporal or precedence-aware planning algorithms not sufficient or applicable here? For example, after BreakLLM decomposes instructions into a formal specification, could a conventional planner (e.g., PDDL-based, temporal constraint solver) replace LocateLLM? If not, what specific limitations (e.g., reasoning over preferences, handling uncertain subgoals, dynamic error correction) prevent that? E.g. see https://ieeexplore.ieee.org/abstract/document/10611163
5. Can the authors elaborate on why and how each component was selected for integration? More discussion of failure cases that motivated the components or justification for the chosen components would be beneficial. This could also be accomplished by more discussion on the ablation study.

**Ethical Concerns:**

["NO or VERY MINOR ethics concerns only"]

**Final Justification:**

The authors have addressed my primary concerns regarding the differentiation between task preferences and detailed task criteria, provided a thorough rationale for component selection, expanded on computational efficiency with planned baseline comparisons, and offered a more comprehensive discussion of why traditional planning algorithms are not applicable in this setting, including relevant citations. While I still view the work as more incremental than groundbreaking in novelty, the benchmark contribution, experimental results, and clarifications provided tip the balance toward acceptance for me.

**Limitations:**

The authors partially address limitations, acknowledging mode switching instability and dependency on pre-trained models. However, explicit discussion of high failure rates and computation overheads would be beneficial. I recommend a dedicated discussion on these points.

**Paper Formatting Concerns:**

No major formatting issues.

**Quality:**

3

**Strengths And Weaknesses:**

Strengths:
- The paper is well-organized, clearly delineating the motivation, methodology, and results.
- The motivation for addressing longer-horizon tasks with multiple subtasks is compelling and effectively articulated.
- The comprehensive engineering effort to integrate multiple modules (LLMs for instruction handling, semantic mapping, policy-based navigation) into a cohesive system is impressive and demonstrates considerable practical innovation.

Weaknesses:
- The emphasis on "task preferences" appears conceptually promising, but operationally resembles detailed task criteria rather than true user preferences, such as choosing among multiple suitable objects based on subjective user criteria.
- The solution seems heavily reliant on the integration of existing components, making the primary contribution more of an engineering achievement than a novel theoretical advancement or the identification of specific problems that inspire the need for different components.
- The paper lacks a deeper discussion on computational efficiency and failure cases, which are critical for real-world applicability given the described "high" failure rates.

Other comments:
There is redundancy in early sections where components are repeatedly introduced. Streamlining these descriptions would allow space for deeper analysis, such as expanded discussion of the ablation studies or detailed insights into the specific challenges addressed by each component.

---

> ### Author Rebuttal · Authors · 2025-07-31
>
> # Response to Reviewer gPWR
>
> We sincerely thank the reviewer for their thorough review and valuable feedback.
>
> ## Response to Weakness 1 \& Question 1: Task Preferences vs. Subtask Criteria
>
> - We strongly agree that a clearer differentiation is crucial. Our "task preferences" address a key gap in prior demand-driven navigation (DDN) datasets, moving beyond object recognition to model abstract user intent.
>
> - Motivation for Task Preferences: Early DDN tasks often featured very simple commands like "I want to sleep". Such instructions are highly ambiguous, as "sleeping" could involve a sofa, a bed, or even resting on a desk, which makes inferring intent and target objects difficult. We propose that each subtask should have an implied or explicit preferred task preference to guide agents.
>
> - Differentiating "Object Preferences" from "Task Preferences":
>
> (1) For a macroscopic instruction like "organize the living space", user preferences can be categorized. "Object preferences" would involve specifying particular objects (e.g., "use a broom", "find a storage box"). However, introducing explicit object information can easily simplify the task to object navigation.
>
> (2) In contrast, our "task preferences" specify the preference of task type, and we ensure that the name of the target object will not be included in the preference. Examples include "achieve cleaning tools", "prepare decorative items," or "find storage tools," as highlighted in lines 23-25 of our main text. This design ensures that the agent must infer the most suitable objects based on the abstract task preference, rather than relying on explicit object names.
>
> (3) Therefore, our "task preferences" are not merely detailed task criteria. The clarity of this requirement is important for correctly executing an abstract basic requirement
>
> ## Response to Weakness 2: Reliance on Existing Components
>
> - We appreciate the reviewer for the thoughtful feedback. We clarify that our core contribution goes beyond combining existing components. It lies in: (1) identifying previously overlooked limitations of current methods in multi-demand navigation, and (2) introducing novel technical designs to effectively address these challenges, enabling robust and efficient performance.
>
> - Challenges and our contributions:
>
> (1) **Map construction**: Previous work directly aggregates all encountered point clouds, which performs well over short distances but incurs high computational overhead and slows down reasoning in long-range exploration. To address this, we propose the Multi-dimensional cumulative Semantic Map (MASMap), a novel contribution designed to efficiently manage spatial semantic memory for long-horizon navigation. Its core innovation lies in sophisticated fusion strategies that leverage Ram-Grounded-SAM for object detection, Euclidean distance for spatial relationships, and IoU for object overlap to extract 3D object point clouds from only local 8-view images. More importantly, our global 2D semantic map explicitly tracks and associates observations across multiple views, effectively preventing redundant mapping and duplicate writes for the same object, thereby ensuring a compact, consistent, and efficient representation essential for scalability in large-scale environments.
>
> (2) **Dynamic Objective Decision-Making**: Current large-model-based demand-driven navigation methods primarily rely on map-point-driven action generation, making them prone to local repetitive planning and lacking dynamic planning awareness for multiple, evolving tasks. To address these challenges, we propose the Autonomous Decision World Model System (AWMSystem), which represents a significant advance in enabling dynamic decision-making without requiring additional training. It integrates an effective chain-of-thought mechanism for object and pose prediction, an adaptive error corrector based on real-time environmental perception to ensure robustness, and a multi-modal large-model-aided task state tracker for more accurate progress monitoring and goal management.
>
> (3) **An Novel Evaluation Task for Multi-Demand-Driven Navigation**, as important as explained in Response to Weakness 1 & Question 1. Our extensive experiments on this benchmark show that our method achieves state-of-the-art (SOTA) performance in both overall task success rate and subtask completion rate.
>
> (4) **Balancing Zero-Shot Transfer Ability and Efficient Reasoning**. As shown in Table 2 (our ablation study), the design choices and individual contributions of the components have a significant impact on navigation performance, demonstrating the essentiality of each component.
>
> ## Response to Weakness 3 \& Question 3: Computational Efficiency and Failure Cases.
>
> ### **Computational Efficiency**
>
> - We agree that a deeper discussion on computational efficiency and failure cases is critical for real-world applicability. The average execution time for a complete task is approximately 17.41 minutes(nearly 39.75 steps).
> - The average time consumption of each module for one task is as follows. BreakLLM :2.89 seconds. Fast-tempo Execution: 67.49 seconds (nearly 32 steps, averaging 2.02 seconds per step). Slow-tempo Execution: 870.74 seconds (nearly steps, averaging 1.81 minutes per step). StatusLLM: 93.93 seconds (averaging 4.68 seconds per call). Adaptive Error Corrector: 12.64 seconds.
> - Smaller modules in slow-tempo execution is as follows. Real-time Mapping: 93.68 seconds. LocateLLM: 5.06 seconds. Affordance: 4.13 seconds. Waypoint Predictor: 0.75 seconds. Trajectory Execution: 5.24 seconds.
> - It takes about 0.39 seconds for a robot to move or rotate once. This detailed breakdown demonstrates how our dual-tempo design balances robustness with efficiency.
>
> ### **Analysis of Failure Cases**
>
> We observed recurring failures related to accuracy and object placement, yet our method still remains viable for real-world use, as shown in our real-robot experiment (see attachment). (1) In cluttered scenes, limited spatial reasoning may cause the LLM to treat nearby objects as separate, leading to redundant visits. However, this occurs significantly less often than with prior map-point-driven methods. We can attempt to enhance the coordinate perception awareness of large models. (2) Small Object Misses: Very small items such as clips or pencils may be missed despite RAM-Grounded-SAM's robustness. Future work could explore zoom-in detection. (3) Hidden Targets: Hidden Targets: If an occluded target is deemed absent, the agent may skip the area, causing failure. This suggests a need for active reasoning, such as checking cabinets, through knowledge bases or uncertainty-driven exploration.
>
> ## Response to Other Comments \& Question 5: Component Selection.
>
> We acknowledge the reviewer's comment regarding adding expanded discussion.
>
> ### **Justification for Component Selection and Integration:**
>
> Our system is modularly designed, comprising map construction, command decomposition, and dual-tempo action generating modules. Each component was carefully selected and designed to address specific limitations of prior work in the context of multi-demand long-range navigation:
>
> **- component :**
>
> 1. Map Construction: Problem: Methods like InstructNav store all point clouds, causing slow reasoning (~1 hour/task). Our Fix: We merge 3D points using panoramic views and anti-repetition compression, enabling scalable, fast mapping.
>
> 2. Target Object Prediction (LocateLLM): Previous work relies on map-point-driven target prediction, often leading to redundant exploration within the same area. In contrast, our LocateLLM performs object-level prediction, significantly reducing the likelihood of repeated visits. Moreover, we innovatively predicts the target coordinates directly, providing precise spatial input that meets the requirements for subsequent trajectory planning.
>
> 3. Adaptive Error Corrector: Despite effective path planning, collisions can still occur in tight spaces. We attempted to use multimodal LLMs for correction, but the action sequences they generated failed to smoothly navigate around obstacles due to a lack of simulation-based planning experience. In contrast, our adaptive corrector leverages real-time environmental feedback to autonomously resolve collisions and boundary issues, significantly improving navigation robustness.
>
> **- Ablation Study Insights (Table 2):**
>
> 1. Segmentation: Grounded-SAM outperformed GLEE (used in DDN/InstructNav), especially in detecting fine details (e.g., cup on toilet)
>
> 2. State Monitoring: Using LocateLLM to simultaneously perform status prediction failed, likely because the multitasking prompt diluted its focus and impaired judgment accuracy. In contrast, our hybrid approach, where the expert module triggers "Done" and the LVLM independently confirms completion, significantly improved success rates.
>
> 3. LLM Comparison: Qwen2.5-VL-7B exhibited repeated targets, while GPT-4o switched subtasks erratically. In contrast Qwen3-32B demonstrated a more reasonable rhythm in subtask switching.
>
> ## Response to Question 4: Traditional Planning Algorithms
>
> Traditional planning algorithms (e.g., PDDL) are unsuitable for replacing LocateLLM because they: (1) focus on temporal or priority constraints, while our method emphasizes interpreting multimodal inputs to fulfill abstract instructions; (2) require discrete, predefined state spaces that conflict with our unseen, continuous 3D environments; (3) struggle with dynamic, abstract sub-goals and adaptive task prioritization based on evolving context; and (4) lack robust real-time error correction from continuous sensory feedback. Unlike approaches such as AutoTAMP that use LLMs for output correction, our method employs LLMs for core chain-of-thought reasoning and multimodal understanding.
>
> ## Details:
>
> Response to Question 2. Subtasks are evaluated as binary outcomes, either successful or failed.

---

> ### Author Response · Authors · 2025-08-06
>
> Dear Reviewer,
>
> I hope this message finds you well. As the discussion period is **nearing its end with less than two days remaining**, I want to kindly check **whether we have addressed all your concerns satisfactorily**. If there are any further points or suggestions you feel we could clarify or improve, we would be truly grateful for your guidance. Your insights are invaluable to us, and we are eager to address any remaining issues to improve our work.
>
> We deeply appreciate the time, care, and thoughtful effort you have devoted to reviewing our manuscript. Thank you once again for your invaluable contribution.
>
> Best regards,

---

> ### Comment · Reviewer_gPWR · 2025-08-06
>
> Thank you for your detailed and thoughtful responses. You’ve addressed my concerns regarding the conceptual differentiation between task preferences and detailed task criteria, provided valuable insights into your component selection and integration choices, and offered an extensive breakdown of computational efficiency and an analysis of failure cases. These clarifications enhance my understanding of your contributions.
>
> To further strengthen the manuscript, I suggest the following minor points for additional clarification:
>
> - The provided computational time breakdown is valuable. Contextualizing these results by directly comparing your method’s efficiency against baselines could further highlight its practical strengths and justify the real-world applicability more convincingly.
>
> - Regarding traditional planning algorithms, your justification effectively notes their limitations in handling multimodal inputs and dynamic adaptability. However, a more explicit comparative reference or supporting citation to prior work demonstrating these limitations would further solidify your argument and provide clearer guidance to the reader.
>
> Overall, your responses have improved my assessment of the paper, and I encourage these minor additions to further enhance clarity and rigor.

---

> > ### Author Response · Authors · 2025-08-07
> >
> > **Response to Reviewer gPWR**
> >
> > We sincerely thank the reviewer for the valuable and insightful comments. We truly appreciate the constructive suggestions, and we will incorporate the points raised into the revised version of our manuscript. Below, we provide some clarifications in response to the feedback.
> >
> > ---
> >
> > ## **Computational Efficiency**
> > Regarding computational efficiency, we will include a more detailed comparison between our method and the baseline approaches. As for the average execution time for a complete task, our method takes approximately 17.41 minutes, whereas other baselines such as InstructNav require around one hour, and DDN and MO-DDN take less than 5 minutes. Our approach achieves a significantly higher success rate compared to existing baselines, while maintaining a reasonable and acceptable inference speed. Additionally, we will provide a breakdown of the inference time for each intermediate component to further highlight the practical advantages of our method.
> >
> > ## **Explicit Comparative Reference**
> > With regard to the citation concerning the limitations of traditional planning algorithms in our task setting, prior works have explored the use of traditional algorithms or LLM-augmented traditional methods for Task and Motion Planning (TAMP) to address robotic trajectory planning. Some approaches directly employ large language models (LLMs) for high-level semantic planning but primarily focus on sub-step planning [1,2], which does not guarantee continuous action execution and thus limits their applicability in real-world navigation tasks. Other studies utilize programming-style prompts to generate plans [3] or incorporate environmental feedback such as task completion signals [4]; however, these are mostly designed for manipulation actions (e.g., pick up, open, close) and are difficult to extend to long-horizon navigation tasks involving abstract instructions and multimodal progress recognition.
> >
> > To handle complex sequential operations or constrained tasks, researchers have proposed geometric feasibility checkers with search-based planning [5], generated plans based on spatial object relations [6], modeled problems in discrete state spaces using PDDL or other AI planning languages [7,8,9], or adopted hierarchical planning for temporal logic constraints [10]. However, these methods typically require predefined, discrete state spaces—conditions that do not hold in our multi-demand navigation setting, where environments are unseen and cannot be fully specified in advance. Moreover, search-based algorithms often struggle to find all feasible trajectories in continuous 3D environments over long-horizon tasks, which may involve dozens of steps. Crucially, unlike the aforementioned traditional planning tasks with clearly defined goals, our task is demand-driven, where the target object is dynamically determined during real-time exploration based on the agent’s current context and needs.
> >
> > [1] Language Models as Zero-Shot Planners: Extracting Actionable Knowledge for Embodied Agents
> > [2] Do As I Can, Not As I Say: Grounding Language in Robotic Affordances
> > [3] PROGPROMPT: Generating Situated Robot Task Plans using Large Language Models
> > [4] Inner monologue: Embodied reasoning through planning with language models
> > [5] Text2motion: From natural language instructions to feasible plans
> > [6] Task and Motion Planning with Large Language Models for Object Rearrangement
> > [7] Combined Task and Motion Planning as Classical AI Planning
> > [8] AutoTAMP: Autoregressive Task and Motion Planning with LLMs as Translators and Checkers
> > [9] Pddlstream: Integrating symbolic planners and blackbox samplers via optimistic adaptive planning
> > [10] Towards Manipulation Planning with Temporal Logic Specifications
> >
> > ---
> > Once again, we are deeply grateful to the reviewer for the thoughtful and helpful feedback. The comments have greatly contributed to improving the clarity and rigor of our work.

---

### Official Review · Reviewer_J2bP · 2025-06-30

**Clarity:** 3
**Significance:** 2
**Originality:** 3
**Rating:** 4
**Confidence:** 3

**Summary:**

This paper introduces a multi-demand object goal navigation benchmark TP-MDDN and an AWMSystem decision-making system, which can achieve SOTA performance on the benchmark. TP-MDDN is constructed based on AI2-THOR simulator on ProcThor scenes, where authors construct 100 task-preferenced multi-demand-driven unseen instructions. The AWMSystem is a complicated system which incorporates MASMap spatial memory scheme, Dual-Tempo action generation framework, and the adaptive error corrector.

**Questions:**

- Line 119 states that the benchmark uses a distance threshold to determine whether the target is found. This is very unclear. What specific distance metric is used—Euclidean distance or geodesic distance? Additionally, should the target be observable by the agent, as in other navigation benchmarks (e.g., Habitat HM3D ObjectNav)?

- The LLM is tightly coupled with the entire system. Does increasing the map size (and thus the number of prompts) lead to a performance drop in the LLM?

- I note that the method supports looking up and looking down, but the real-world experiments do not include these actions. Is there any discussion about this discrepancy?

**Ethical Concerns:**

["NO or VERY MINOR ethics concerns only"]

**Final Justification:**

Thanks for the thoughtful discussion. Most of my concerns have been addressed.

I agree that this paper on demand-driven navigation is interesting and the whole work required large efforts, but my remaining concern is the heuristic design of using LLM, which leads to a lack of sufficient novelty. Overall, I maintain my initial score.

**Limitations:**

Yes (Section 5, more detailed discussion will be appreciated.)

**Paper Formatting Concerns:**

No paper formatting concerns

**Quality:**

3

**Strengths And Weaknesses:**

Strengths: This paper is highly comprehensive, with significant contributions to both benchmarks and methods. I particularly appreciate the illustrations and the video, which effectively demonstrate the methodology and performance. The authors’ ability to make such a complex method work in both simulated and real-world environments is commendable.

Weaknesses:

- Justification for demand-driven vs. object-driven navigation: The pipeline attempts to decode instructions into a series of target objects and then perform object-goal navigation. This raises a key question: If the LLM is already effective at translating demands into target objects, why is a separate demand-driven navigation method necessary? This task could potentially be handled by object-goal navigation methods, which are well-established in the field.

- Heuristic-based pipeline relying on LLMs and maps: While a heuristic-based pipeline that performs well is acceptable, the paper lacks methodological novelty and key insights. Authors could highlight the key technical contribution of this paper.

---

> ### Author Rebuttal · Authors · 2025-07-31
>
> # Response to Reviewer J2bP
>
> We sincerely thank the reviewer for their thorough review and valuable feedback.
>
> ## Response to Weakness 1: Justification for Demand-Driven vs. Object-Driven Navigation.
>
> We'd like to clarify that our approach does not directly decode instructions into fixed target objects at the outset. Instead, our demand-driven framework addresses several critical challenges that go beyond the scope of traditional object-goal navigation:
>
> - Dynamic and Abstract Target Selection: Unlike object-goal navigation, where a specific object is the fixed target, our system first identifies objects and their positions from the scene. Then, a large language model (LLM) dynamically determines which sub-task to execute currently and which object to select as the target based on the abstract instruction and the evolving environment. This process also critically involves ensuring that previously selected objects are not duplicated. This dynamic, context-aware decision-making for abstract instructions is a core challenge of demand-driven navigation that object-goal navigation cannot directly resolve.
>
> - Robustness to Object Recognition Uncertainty: In real-world and simulated environments, object recognition is not 100\% accurate due to factors like lighting, shadows, and color distortion. Even with advanced segmentation models like RAM-Grounded-SAM, the perceived names of the same object from different perspectives can be inconsistent or inaccurate. Our system accounts for this by requiring the agent to explore and verify whether a recognized object truly meets the abstract demand, rather than blindly navigating to a pre-defined object.
>
> - Real-time Adaptation to Unknown Scenes: We cannot determine the ideal target object for abstract requirements in an unknown scene from the very beginning. Our system continuously records newly detected objects in real-time and feeds them as optional targets to LocateLLM. This means that for the same abstract demand, the LLM dynamically adjusts its target object selection based on newly recognized objects during exploration, enabling adaptive behavior in unseen environments.
>
> Therefore, our demand-driven navigation method is essential, which are fundamentally distinct from and more complex than simple object-goal navigation.
>
> ## Response to Weakness 2: Methodological Novelty.
>
> We thank the reviewer for their detailed assessment. We respectfully clarify that our primary contribution is not merely an engineering achievement. Instead, it lies in identifying specific, unresolved problems within the context of the novel multi-demand-driven navigation task and proposing innovative technical designs to overcome these challenges. This constitutes significant methodological novelty. Our key technical contributions are:
>
> - **Efficient and Scalable Spatial Memory Constructing in long-horizon navigation**. Previous work directly aggregates all encountered point clouds, which performs well over short distances but incurs high computational overhead and slows down reasoning in long-range exploration. To address this, we propose the Multi-dimensional cumulative Semantic Map (MASMap), a novel contribution designed to efficiently manage spatial semantic memory for long-horizon navigation. Its core innovation lies in sophisticated fusion strategies that leverage Ram-Grounded-SAM for object detection, Euclidean distance for spatial relationships, and IoU for object overlap to extract 3D object point clouds from only local 8-view images. More importantly, our global 2D semantic map explicitly tracks and associates observations across multiple views, effectively preventing redundant mapping and duplicate writes for the same object, thereby ensuring a compact, consistent, and efficient representation essential for scalability in large-scale environments.
>
> - **Dynamic Objective Decision-Making for Multiple Needs**: Current large-model-based demand-driven navigation methods primarily rely on map-point-driven action generation, making them prone to local repetitive planning and lacking dynamic planning awareness for multiple, evolving tasks. To address these challenges, we propose the Autonomous Decision World Model System (AWMSystem), which represents a significant advance in enabling dynamic decision-making without requiring additional training. It integrates an effective chain-of-thought mechanism for object and pose prediction, an interaction mechanism based on environmental real-time perception, and a multi-modal large-model-aided task state tracker for more accurate progress monitoring and goal management.
>
> - **High-Precision Motion Generation for Robust Navigation**: Nowadays, demand-driven action generating methods include using large models to predict the the current one action. However, agents are prone to being attracted by multiple objects in the scene and deviating from the previously planned path, causing decision-making confusion. Some methods approach the detected related objects, but they are often difficult to apply in unseen scenes due to urgent problems such as being unable to pass through narrow doors and getting stuck at table corners. To fill this gap and address issues such as door frame obstruction and exceeding boundaries without additional training, we propose a waypoint predictor targeting the center coordinates of objects, with multi-size collaborative correspondence (agent step size, object size, waypoint sampling interval), and then use an adaptive action corrector based on environmental feedback. Ultimately, in the multi-demand navigation task, we achieved a trade-off between success rate and reasoning performance by using the Dual-Tempo action generation method with efficient approximation of the target object and fine expert detection.
>
> ## Questions
>
> ### Response to Question 1: Distance Metric.
>
> - Thank you for the question. To clarify, we define successful sub-task completion using the 2D Euclidean distance between the agent and the center of the target object, with a success threshold of 1.5 units. While benchmarks such as Habitat's HM3D ObjectNav require visual observation of the target object, we do not explicitly verify such observation during evaluation. This choice was made to reduce complexity in the context of multi-demand driven navigation, as discussed in the main text (lines 118–122). A consistent evaluation protocol was applied across all baseline experiments. Furthermore, extensive visual comparisons demonstrate that our method achieves strong performance in accurate target exploration, effective obstacle avoidance, and the resolution of complex multi-demand tasks.
>
> ### Response to Question 2: LLM Coupling with Map Size / Performance Drop
>
> - We confirm that the size of the semantic map does not directly affect the number of LLM prompts or lead to a decline in LLM performance. This is because our prompt generation is object-based rather than map-based. we detect objects within the agent’s current field of view, compute their 3D coordinates in the camera coordinate system, and then transform them into the real-world coordinate system based on the current camera pose. Only the names and center coordinates of these detected objects are included in the prompt, meaning the number of prompts depends solely on the visible objects at any given moment, not on the overall map size. Furthermore, our 2D semantic grid map—constructed by extracting and merging 3D point clouds from panoramic RGB-D images (captured from 8 perspectives), projecting them onto a 2D plane, and compressing them into a pixel grid—serves primarily as lightweight spatial memory, increasing storage by only kilobytes to megabytes. Crucially, this map is not fed directly into the LLM as input, and therefore does not contribute to prompt length or impact LLM performance.
>
> ### Response to Question 3: Look Up/Down in Real-World Experiments
>
> - We acknowledge that, although our method supports "look up" and "look down" actions, these are not explicitly included in the real-world experiments. This limitation arises from two main factors: first, the camera on our real robot is mounted along the arc of the mechanical arm, which imposes mechanical constraints that make precise control of vertical gaze directions challenging. Second, the tasks in our real-world scenarios primarily involve common daily-life objects, which can be effectively observed from the robot’s standard horizontal (forward-facing) camera perspective. As such, the absence of explicit vertical viewpoint changes does not significantly hinder the robot’s ability to perceive relevant objects in the environment.

---

> > ### Comment · Reviewer_J2bP · 2025-08-05
> >
> > Thanks for the thoughtful discussion. Most of my concerns have been addressed.
> >
> > I agree that this paper on demand-driven navigation is interesting and the whole work required large efforts, but my remaining concern is the heuristic design of using LLM, which leads to a lack of sufficient novelty. Overall, I maintain my initial score.

---

> > > ### Author Response · Authors · 2025-08-05
> > >
> > > **Response to Reviewer J2bP**
> > >
> > > We sincerely thank the reviewer for the time and valuable feedback throughout the review process.
> > >
> > > - Our work focuses on the newly introduced multi-demand-driven navigation task, where leveraging LLMs is not a straightforward plug-and-play solution. Despite their strong language understanding, current LLM-based methods still face significant challenges in vision-and-language navigation, particularly in grounding, planning, and robust interaction, often resulting in low success rates. Our approach integrates LLMs into a structured framework—AWMSystem—with specific designs for task decomposition, goal selection, and progress monitoring, enabling effective decision-making without requiring large-scale multi-task training data.
> > >
> > > - This is the first work to address long-horizon navigation with multiple demands. While end-to-end expert models are a promising direction, they entail substantial data and training efforts, which we leave for future exploration.
> > >
> > > Should the reviewer have any further questions, we would be happy to provide additional clarification. Thank you again for the thoughtful comments.

---

### Official Review · Reviewer_DyKS · 2025-07-01

**Clarity:** 3
**Significance:** 2
**Originality:** 2
**Rating:** 4
**Confidence:** 5

**Summary:**

This paper introduces Task-Preferenced Multi-Demand-Driven Navigation (TP-MDDN), a new benchmark for long-horizon navigation involving multiple sub-demands with explicit user preferences, addressing a key gap in existing research. To solve this, the authors propose the Autonomous Decision-Making World Model System (AWMSystem). This system uses three core LLM modules for reasoning: BreakLLM for decomposing instructions, LocateLLM for selecting goals, and StatusLLM for monitoring task completion. For spatial memory, it employs MASMap, which efficiently integrates 3D point clouds into a 2D semantic map. A key innovation is the Dual-Tempo action generator, which combines slower, high-level planning with fast, policy-based control to enhance efficiency and navigation robustness. An Adaptive Error Corrector further improves reliability by handling real-time navigation failures. Experimental results show the system significantly outperforms state-of-the-art baselines on the TP-MDDN benchmark

**Questions:**

1. Did the authors use the same segmentation model for the baselines? Because the performance of the segmentation model like in InstructNav can very much affect the navigation performance. Using the same segmentation model for all baselines is a fair comparison that demonstrates systematic superiority.

**Ethical Concerns:**

["NO or VERY MINOR ethics concerns only"]

**Final Justification:**

The author clarifies the difference with the previous paper (DDN and MODDN) very well.

**Limitations:**

yes

**Quality:**

2

**Strengths And Weaknesses:**

## Strengths

1. The method proposed in the paper is a very well developed modular system with very good scalability. The fast-slow system is a good solution to the problems of slow inference for large models and poor generalization of end-to-end models.
2. The entire system is designed to be robust and safe with an Adaptive Error Corrector.
3. Overall, the method designed by the authors is effective and exceeds the baseline in the experimental results.

## Weaknesses

1.  I feel a little bit strange, why merge several unrelated demands into one instruction? For example, in Fig 3, "Organize sports equipment storage, set up a beverage station with drinkware, and maintain bathroom cleanliness with hygiene products".  In a real-world environment, it doesn't seem to be a bother for a user to give several individual demand instructions. In addition, it does not seem to be a difficult task for LLM to break down these independently demand instructions nowadays. I think the author needs to explain the motivation for designing this benchmark
2. There are some errors in the author's description of the MO-DDN near lines 33 to 35. The MO-DDN does not emphasize object names; like the DDN, it essentially satisfies a demand, so as long as an object satisfies a demand it does not require a specific object. "Organize lunch" implies that the agent needs to find the food first and then find the cooking tools , which is the meaning of its prefix MO (Multi-Object). That's because "organize lunch" can't be accomplished without finding food or without finding cooking tools.
3. While the authors have designed an approach that exceeds the baseline, I feel there is some lack of novelty. As an example, Construction of long-range memory banks and affordance maps are used in InstructNav, and it is not the first time that the Dual-Tempo action generator has been used in navigation. I suggest that the authors elaborate on the differences and re-explain the novelties in their paper.

---

> ### Author Rebuttal · Authors · 2025-07-31
>
> # Response to Reviewer DyKS
>
> We sincerely thank the reviewer for the insightful and constructive feedback.
>
> ## Response to Weakness 1: Motivation for Merging Unrelated Demands.
>
>  - Thank you for the valuable feedback. We designed this multi-demand benchmark to evaluate and advance embodied AI in multi-task, long-range navigation. Our design considerations fall into two main aspects. First, we aim to enhance agents' multi-task navigation capabilities, as current single-task navigation accuracy is still limited, and multi-task scenarios tend to accumulate errors. Second, given the limited variety of object types in existing simulation environments—which constrains the creation of multiple related sub-tasks under a single overarching goal—we have instead constructed unrelated sub-tasks. This approach is a pioneering exploration in addressing multi-unrelated-demand-driven navigation challenges.
>
> Here are more specific explanations:
>
> - Addressing Cumulative Error in Sequential Execution: Current single-demand navigation methods often exhibit low success rates. For example, even a strong baseline like InstructNav achieves only 30.0% success on the DDN dataset for single demand. When multiple unrelated demands are executed sequentially, the errors accumulate, leading to a significantly lower overall success rate.
>
> - Bridging the Gap in Multi-Demand Paradigms: We identify a notable gap in existing demand-driven navigation methods for their limited performance in handling multi-demand tasks. While some approaches, such as MO-DDN, suggest potential for multi-step, related requirements (as indicated in its teaser), their datasets and prompts typically focus on fulfilling single preference with high precision. In contrast, our work proposes a novel framework specifically designed to improve success rates on instructions with several high-level demands that do not explicitly name the required objects.
>
> - **Comparison with experiment that executes single tasks multiple times**. Verifiably, we also proved through experiments that the success rate of the DDN method in successively and separately executing each sub-task is much lower than that of our method. Their results are **SR=16\% and ISR=47.74\%.** This indicates that the method of splitting instructions into several individual demand instructions for sequential execution with the aid of large models does not lead to a significant increase in the success rate of multi-demand instruction navigation. It is still very necessary to propose innovative methods to enable agents to efficiently solve multi-demand long-term tasks.
>
> ## Response to Weakness 2: Clarification on MO-DDN Description.
>
> - We sincerely appreciate the reviewer's valuable clarification regarding the nature of MO-DDN. We fully agree with the reviewer's understanding that MO-DDN is indeed a demand-driven navigation method. To clarify our original statement,  the "emphasizes object names" we mentioned in Line 32 of the main text actually refers to the Task Dataset Generation prompt and the example it generates, detailed in Section A.1.1 of the MO-DDN attachment. In that section,  the preference requirements for generating raw tasks in their prompt contain "**Detail any additional preferences that refine the object selection**", which may introduce actual object information and simplify the instructions. For instance, just as their examples on page 18 prompts like "**I hope the bed is comfortable enough**" explicitly include an object name ("bed") alongside a preference. **In contrast**, our evaluation set is designed with a stricter requirement: instructions must not contain the name of the target object, and instead express abstract requirements and include task-preferenced preference. It should be noted that the toys in the subtask "Organize a cozy play area with toys" in Figure 2 of our main text are not our target objects. This design ensures the agent to infer the most suitable objects or locations based on abstract demands and highlights one of key difference compared to MO-DDN.
>
> - We will revise lines 33-35 in the manuscript to ensure this distinction is more accurately and clearly communicated, preventing any misinterpretations. Thank you very much for your suggestion.
>
> ## Response to Weakness 3: Novelty and Differences from Related Work.
>
> Our novelty lies not merely in introducing entirely new concepts, but in the innovative integration, significant refinement, and novel application of these concepts within a holistic framework.
>
> **- Differences from Related Work**
>
> 1. Comparison with InstructNav
>
> **(a) Long-range memory banks**. InstructNav directly aggregates all encountered point clouds. This approach works well for short distances, as shown by its average trajectory length of 4.44, but causes high computational overhead and slows down reasoning in long-range exploration. Our approach innovatively proposes the Multi-dimensional cumulative Semantic Map (MASMap) construction method. This method creates a highly efficient 2D semantic graph by intelligently merging 3D point clouds from a local panoramic perspective, crucially avoiding duplicate perspective records in the global scope. This design significantly enhances the efficiency and scalability of semantic mapping for truly long-range navigation, a critical distinction from InstructNav's more direct aggregation.
>
> **(b) Affordance maps**. Instructnav's affordance graph integrates multiple heterogeneous affordance values (e.g., planning direction, action feasibility, landmarks). This fusion process may distort the spatial relevance, causing the highest-scoring location to deviate from the vicinity of the most semantically relevant object in the current context. Furthermore, since instructnav takes the points with the largest afforsibility value as the target, given the high density of points, this method often results in consecutive points being chosen from the same local region, trapping the agent in repetitive, small-scale movements without effective exploration. Therefore, we propose a trajectory planning method that uses an affordance map for precise pathfinding directly to the target object's location, combined with an adaptive error corrector to enhance planning effectiveness and avoid local oscillations.
>
> 2. Explanation of Dual-Tempo Action Generation
>
> Thank you very much for your insights. Our Dual-Tempo Action Generator is specifically designed to address the inherent trade-off between the meticulous, but slow, planning capabilities of large models and the rapid execution of expert strategies in the context of long-range embodied navigation. We explicitly highlight this distinction by demonstrating a stark difference in planning times: the large model planning takes 1.81 minutes in one step, whereas the fast-process expert strategy executes in just 2.02 seconds in one step.
>
> **- Core Novelties of Our Paper**
>
> - Multi-dimensional cumulative Semantic Map (MASMap): This is a novel contribution specifically designed to tackle the challenge of efficiently managing spatial semantic memory in long-range navigation. Its core innovation lies in its sophisticated fusion strategies, leveraging Ram-Grounded-SAM for object detection, Euclidean distance for spatial relationships, and IoU for object overlap, to extract 3D point clouds of objects from only local 8-view images. More importantly, our global 2D semantic map explicitly tracks and associates observations across multiple views, effectively preventing redundant mapping and duplicate write operations for the same object, ensuring a compact, consistent, and efficient semantic representation crucial for scalability in vast environments.
>
> - Autonomous Decision World Model System (AWMSystem): This system represents a significant step towards dynamic objective decision-making in multi-demand tasks without requiring additional training, which integrates an effective chain of thought for object/pose prediction, an adaptive error corrector based on dynamic environmental perception for real-time robustness, and a multi-modal large model-aided task state tracker for more accurate progress and goal management.
>
> - An Novel Evaluation Task for Multi-Demand-Driven Navigation. This benchmark introduces subtasks that deliberately omit explicit object names, express abstract requirements, and incorporate task-type preferences, challenging agents to move beyond simple object retrieval and demonstrate genuine understanding of complex, real-world demands. Our extensive experiments on this benchmark show that our method achieves state-of-the-art (SOTA) performance in both overall task success rate and subtask completion rate.
>
> # Response to Question 1
>
> - We are grateful for the reviewer's valuable suggestions. In our baselines comparison, the original DDN method does not require a segmentation model, and MO-DDN uses the same segmentation model as us. As for instructnav, to make a fairer comparison, we conducted a set of experimental results after using the same segmentation model as ours by using RAM-Grounded-SAM, and InstructNav achieved a **14\% SR (Success Rate), 3.34 STL (Success weighted by Trajectory Length), 39.39\% ISR (Independent Success Rate), and 20.80\% ISPL (Independent Success weighted by path Length)**. These results highlight that even with identical segmentation capabilities, InstructNav's performance remains significantly inferior to ours, demonstrating our framework's superior efficacy and robustness in complex navigation tasks.

---

> > ### Comment · Reviewer_DyKS · 2025-08-05
> >
> > Thanks for the authors' responses. I have the following concerns.
> >
> > ## Re Response to Weakness 1:
> >
> > I recognize that running multiple demand navigations within the same scenario is indeed a worthwhile direction for research. It makes sense as it involves capabilities similar to lifelong learning, memorization, etc.
> >
> > The point I'm trying to make is that simply putting together multiple separate demands is not a difficult thing to do. In DDN, there is already an instruction set for a demand that corresponds to an object, so why not just stitch together the instructions in DDN?
> >
> >
> > ## Re Response to Weakness 2:
> >
> > As far as I konw, most of the instructions in MO-DDN don't include the concrete name of the target object.
> > The full version of the instruction as noted by the authors is as follows, divided into base and preference demands according to the MO-DDN setting, with the bed being part of the preference demands.
> >
> > > I need a comfortable sleeping arrangement for a guest staying over one night, I hope the bed is comfortable enough.
> >
> > Since the MO-DDN instruction set is publicly available, I suggest that the authors calculate an objective metric for comparing the percentage of specific objects included in the instructions of the MO-DDN (for both base and preferred instructions) and the instructions in this paper.

---

> > > ### Author Response · Authors · 2025-08-06
> > >
> > > **Response to Reviewer DyKS**
> > >
> > > We would like to express our deep gratitude to the reviewer for the invaluable comments and thoughtful guidance throughout the review process. We sincerely apologize for any misunderstandings that may have arisen in our previous responses.
> > >
> > > # **Response to Weakness 1:**
> > >
> > > We truly appreciate the suggestion regarding the use of instructions from DDN. However, after careful consideration, we found several limitations in directly adopting or stitching together DDN instructions for our specific task, which led us to generate multi-demand instructions independently:
> > >
> > > - **Task Ambiguity in DDN Instructions**: While we were indeed inspired by demand-driven navigation (DDN), we observed that some DDN instructions (e.g., "I need some entertainment" or "I need a workspace") express high-level, abstract needs. Such prompts can correspond to multiple possible tasks (e.g., "entertainment" could mean playing music, reading, or exercising), making the task boundary ambiguous even for humans. To ensure clear and well-defined subtasks, we required each sub-demand to specify a preferred task type and generated new instructions accordingly, rather than simply combining existing ones.
> > >
> > > - **Instruction Complexity and Naturalness**: DDN instructions typically follow a simple pattern (e.g., "I want..." or "I need..."), which, when concatenated, result in unnatural and overly mechanical multi-demand prompts. To enhance realism and linguistic diversity, we used large language models to generate coherent, fluent, and naturally phrased multi-demand instructions—such as: "Create a welcoming entryway by arranging decorative items, set up a functional kitchen space with cooking essentials, and organize a relaxing reading nook with comfortable seating and literature.".
> > >
> > > - **Feasibility and Subtask Uniqueness Constraints**: When generating multi-demand instructions, we enforced two key rules: (a) instructions must only be assigned to rooms that contain the required objects; and (b) objects satisfying different subtasks must not overlap, to avoid simplifying the multi-demand task. Under these constraints, we attempted to combine DDN single-demand instructions but found it difficult to generate a sufficient number of valid, distinct multi-demand tasks. Hence, we opted for a custom instruction generation approach.
> > >
> > > # **Response to Weakness 2:**
> > >
> > > - We are deeply grateful for the reviewer's detailed feedback on MO-DDN. We fully agree that most instructions in MO-DDN do not include specific object names, and they are intentionally designed to express high-level needs rather than specify exact objects. Regarding the objective metric for assessing object inclusion in instructions, we employed both GPT-4o and Qwen3, followed by rigorous manual verification, using the following protocol: (1) If an instruction partially mentioned a required object, it was flagged for manual review. (2) Only instructions containing no explicit required object names were considered valid. (3) All evaluation reasoning was logged and manually confirmed.
> > >
> > > - The results confirmed that **our instructions contain no explicit required object names. In MO-DDN, only one instruction (as noted in our rebuttal)**: "I need a comfortable sleeping arrangement for a guest staying over one night, I hope the bed is comfortable enough."—contains a potential reference. Another instruction mentioning "light" in the context which is included in the desired object was deemed not to violate the principle, as "light" refers to a function rather than a specific object.
> > >
> > > - In summary, we hold MO-DDN in high regard for its innovative design in demand-driven navigation. Our work, however, addresses a different problem: Task-Preferenced Multi-Demand Navigation without requiring additional model training.
> > >
> > > Once again, we sincerely thank the reviewer for the time, patience, and insightful comments. We are more than happy to provide any further clarification if needed.

---

> > > > ### Comment · Reviewer_DyKS · 2025-08-06
> > > >
> > > > Thanks to the authors for their replies.
> > > >
> > > > The authors provided an objective metric to compare DDN, MO-DDN and TPMDDN, which addresses my concerns.
> > > >
> > > > I would like the authors to add these comparisons used to enhance the motivation of the task presented in the paper.
> > > >
> > > > I'll raise my score.

---

> > > > > ### Author Response · Authors · 2025-08-07
> > > > >
> > > > > ## **Response to Reviewer DyKS**
> > > > > - We sincerely thank the reviewer for the positive and encouraging feedback. We greatly appreciate the time and thoughtful consideration given to our responses.
> > > > >
> > > > > - We are glad that the clarification regarding the objective metric has addressed the reviewer’s concerns. As suggested, we will incorporate the comparative analysis between DDN, MO-DDN, and our method into the revised manuscript to better strengthen the motivation for the proposed task. Additionally, we will clearly explain the motivation for merging unrelated demands, clarify the respective contributions of related works such as MO-DDN and their distinctions from our work, and add in the experiment section the baselines using the same object segmentation model, along with other valuable suggestions raised during the review period.
> > > > >
> > > > > - We are truly grateful for the reviewer’s constructive comments and the willingness to raise the score. These suggestions have been instrumental in refining and strengthening the revised manuscript.

---

### Decision · Program_Chairs · 2025-09-17

**Decision:**

Accept (poster)

**Comment:**

This paper presents a novel benchmark and system for long-horizon navigation with multiple user sub-demands. The problem is relevant and important, addressing a clear research gap, and the proposed solution is technically sound. Post-rebuttal, concerns were resolved, and reviewers reached consensus for acceptance. The AC concurred and recommended acceptance.